# Doping Effect on Cu_2_Se Thermoelectric Performance: A Review

**DOI:** 10.3390/ma13245704

**Published:** 2020-12-14

**Authors:** Yuanhao Qin, Liangliang Yang, Jiangtao Wei, Shuqi Yang, Mingliang Zhang, Xiaodong Wang, Fuhua Yang

**Affiliations:** 1College of Microelectronics and Research Center of Materials and Optoelectronics, University of Chinese Academy of Sciences, Beijing 100049, China; qinyuanhao17@mails.ucas.ac.cn (Y.Q.); yangliangliang@semi.ac.cn (L.Y.); zhangml@semi.ac.cn (M.Z.); fhyang@semi.ac.cn (F.Y.); 2Engineering Research Center for Semiconductor Integrated Technology, Institute of Semiconductors, Chinese Academy of Sciences, Beijing 100083, China; weijt@semi.ac.cn; 3Electrical, Computer, and Systems Engineering Department, Rensselaer Polytechnic Institute Troy, New York, NY 12180, USA; yangs17@rpi.edu; 4Beijing Institute of Quantum Information Science, Beijing 100193, China; 5Beijing Semiconductor Micro/Nano Integrated Engineering Technology Research Center, Beijing 100083, China

**Keywords:** Cu_2_Se, thermoelectric performance, thermal conductivity, electrical conductivity, Seebeck effect, doping effect, ZT

## Abstract

Cu_2_Se, owing to its intrinsic excellent thermoelectric (TE) performance emerging from the peculiar nature of “liquid-like” Cu^+^ ions, has been regarded as one of the most promising thermoelectric materials recently. However, the commercial use is still something far from reach unless effective approaches can be applied to further increase the figure of merit (ZT) of Cu_2_Se, and doping has shown wide development prospect. Until now, the highest ZT value of 2.62 has been achieved in Al doped samples, which is twice as much as the original pure Cu_2_Se. Herein, various doping elements from all main groups and some transitional groups that have been used as dopants in enhancing the TE performance of Cu_2_Se are summarized, and the mechanisms of TE performance enhancement are analyzed. In addition, points of great concern for further enhancing the TE performance of doped Cu_2_Se are proposed.

## 1. Introduction

Energy conservation is always in the limelight all across the globe due to the current situation whereby limited and unrecoverable fossil fuel energies, like petroleum and coal, take up the main part of human energy consumption. However, with the explosion of human population, energy crisis can only become more and more severe, except recoverable and sustainable energy can be developed and stably utilized to a large extent. According to the international energy outlook of 2019, the U.S Energy Information Administration predicts that energy consumption will rise about 50% from 2018 to 2050, in which the increment contributed by non-OECD (Organization of Economic Cooperation and Development) countries is incredibly close to 70% [1]. It’s even more upsetting that though the energy consumption is quite huge, the efficiency is relatively low, which is around 40% on average [2]. Most of the energy is emitted as heat due to the low utilization efficiency. Not only is the energy resource wasted, but the environment is polluted at the same time, owing to the burning of fossil fuels. In this case, thermoelectric (TE) materials and devices with the property of converting heat to electricity show great promise in energy conservation and environmental protection. In addition, thermoelectric materials and devices also contain peculiar advantages, like no noise, no vibration, high reliability, high stability, flexible size and etc. [3], which make them suitable for widespread of applications [4,5,6].

The TE performance can be described by a dimensionless figure of merit (*ZT*) shown below
(1)$$ZT=\frac{S^{2}\sigma }{\kappa }T$$
in which *T* is temperature with unit K, σ is electrical conductivity, and κ is thermal conductivity, *S* is Seebeck coefficient. Thermal conductivity for semiconductors can be divided into two parts: thermal conduction by carriers and thermal conduction by lattice vibrations, namely phonons. Conduction by carriers follows the Wiedemann–Franz–Lorenz law, which can be described as follows:(2)$$\kappa_{e}=L\sigma T$$ where L is Lorenz number, *T* is temperature, and σ sigma is electrical conductivity. This equation is valid at temperatures close to room temperature or above, at low temperatures this law fails due to the inelastic scattering of carriers. Lattice thermal conductivity can be estimated using simple kinetic theory:(3)$$\kappa_{l}=\frac{1}{3}C_{L}vl$$
in which, $C_{L}$ is specific heat per unit volume, v is phonon velocity, l is phonon mean free path. This equation is very useful in estimating the order of magnitude of lattice thermal conductivity. For semiconductors, in which carriers and phonons both have contribution to thermal conduction, the total thermal conductivity can be written in the form of:(4)$$\kappa =\kappa_{e}+\kappa_{l}$$

Though written in a pretty brief form, thermal conductivity actually cannot be calculated exactly, but can be estimated using perturbation theory and Boltzmann equation. (Specific derivation of equations above and further estimation of thermal conductivity of various kinds of materials can be found in the textbook written by G. S. Nolas et al. [7]). The electrical conductivity can be derived using Drude model, and the final expression is [8]:(5)$$\sigma =n\mu e=\frac{ne^{2}\tau }{m^{*}}$$
where $m^{*}$ is carrier effective mass, *e* is quantity of electric charge, *n* is carrier concentration, μ is carrier mobility and τ is carrier mean free time. The Seebeck coefficient is given by Mott and Jones equation, and if the free electron model is applied, it can be described as [8]:(6)$$S=\frac{8\pi^{2}k_{B}^{2}}{3eh^{2}}m^{*}T{\left(\frac{\pi }{3n}\right)}^{2/3}$$
where $k_{B}$ is Boltzmann constant and *h* is Plunk’s constant. The figure of merit (ZT) has a relationship with the conversion efficiency, given by:(7)$$\eta =\frac{T_{h}-T_{c}}{T_{h}}\frac{\sqrt{1+\mathrm{ZT}}-1}{\sqrt{1+\mathrm{ZT}}+T_{c}/T_{h}}$$
where $T_{c}$ is the temperature of the cold end, $T_{h}$ is the temperature of the hot end, ZT is the figure of merit. From Equation (5), $\frac{T_{h}-T_{c}}{T_{h}}$ is also refer to as the Carnot efficiency [9], describing the highest efficiency that a heat engine can reach. Usually, an efficient TE device should have a ZT ≥ 3, in other words it should have a efficiency around 20–25%, which is about half of the Carnot efficiency [10]. At this stage, the thermoelectric device can be commercially feasible [11].

From Equation (1), ZT has a strong relationship with temperature T, thus, according to the optimum operating temperature, the thermoelectric materials are divided into three groups, which are: low temperature (<400 K) TE materials: Bi_2_Te_3_ series; mid temperature (600−900 K) TE materials represented by PbTe series; high temperature (>900 K) TE materials which are represented by SiGe series. These series of materials have already achieved high figure of merit (ZT) values, and are applied in some certain field [12,13,14,15,16]. For instance, SiGe and PbTe series of TE materials, using radioisotope as heat source, have been used in space domains. TE device, driven by the radioisotopes (plutonium metal 238 specifically), works as power supplies for probes travelling far away from the Sun, where low intensity of light makes the solar panels fail to work [17,18]. Another example would be applications in automobile industry, where SiGe and Bi_2_Te_3_ series of TE materials have been used to turn waste heat to electrical power. The TE generator, composed of 16 B_i2_Te_3_ modules, can generate electricity with a power of 193 W [19]. What’s more, TE materials can also exploit heat generated by various of resources, like, human body [20,21], solar energy [22] and etc. However, it can be noticed that those series of materials, i.e., intermetallic materials of elements like Pb, Te, Bi, and etc., are typically scarce and expensive. More importantly, they contain toxic heavy metals and are readily to cause environmental pollution [23,24]. It will inolve a higher risk of severe pollution to put them into widespread commercial use. Thus, looking for TE materials which are environmentally friendly and have high efficiency becomes a buzz in recent TE materials researches [25,26]. Cu_2_Se is one of the promising materials.

Cu_2_Se is a p-type semiconductor. The bandgap of this semiconductor can be determined through ab initio calculation, but the results varies using different functional. The most recent result given by Sahib Hasan et al., using Perdew–Burke–Ernzerhof (PBE) generalized-gradient approximation (GGA), is 0.53 eV [27], and bandgaps around 0.5 eV is also presented by many other studies using different approximations and functionals [28,29]. In year 2012, a concept known as “phonon-liquid” was proposed by Huili Liu et al. [30]. Different from the “phonon-glass” put forward by Slack in 1976 [31] (which is a minimum glass-like thermal conductivity can be reached in crystals with long-range atomic order, but a disordered atomic arrangement), this “phonon-liquid” phenomenon discovered in copper selenide prohibits heat propagation through transverse, shear vibration that still happens in “phonon-glass” materials, i.e., the copper selenide possess an ultralow thermal conductivity like a liquid, further contributing to an enhanced ZT value of 1.5 at 1000 K. This liquid like behavior of copper selenide, stems from its crystal structure. From the phase diagram shown in Figure 1a, Cu_2_Se displayed a phase transition at temperature around 396 K [32,33]. At temperatures below 396 K, Cu_2_Se is in α-phase, and at temperature above 396 K, Cu_2_Se is in β-phase. In α-phase Cu_2_Se, Se atoms stacks in close packed pattern with ABC layer-sequence, and Cu atoms are placed at tetrahedral interstices as presented in the schematics in Figure 1b [34], however, the crystal structure is uncertain: monoclinic, tetragonal and cubic are all equally possible [35]. And in β-phased Cu_2_Se, Se^2−^ ions are still in closed-packed arrangement, Cu^+^ ions now will have a superionic transport property (a very high diffusion coefficient about $3.4\times 10^{-5}{\;\mathrm{cm}}^{2}s^{-1}$), which is the very origin of the liquid like behavior of Cu^+^ ions [27]. Lubomir Gulay et al. used an FCC unit cell to describe the crystal structure, as displayed in Figure 1c. This time, as shown in the crystal structure, Cu atoms have 3 possible ways of placement: 0.31 Cu1, 4 × 0.14 Cu2 and 4 × 0.03 Cu3 with total occupancy of ~1 [34]. This model can also be supported by the model given by S.A. Danilkin et al., through neutron scattering study, which is Cu^+^ occupying 8c (0.25, 0.25, 0.25) and 32f (x, x, x) (x = 0.33–0.39) sites [36]. This special crystal structure at At high temperature means Cu^+^ ions are easily drifted within the tetrahedral interstices among those 3 sites, forming a liquid behavior of Cu^+^ ions, which like a liquid, and the trajectories of Cu^+^ and Se^2−^ ions can be illustrated in Figure 1d, the ab initio molecular dynamics (AIMD) simulations [37], in which Cu+ ions are highly disordered and Se^2−^ ions form a relatively rigid framework. Similar result is also obtained in the AIMD simulation on trajectories of individual Cu^+^ ions by Keenan Zhuo et al. [27]. This special crystal structure at high temperature means Cu^+^ ions are easily drifted within the tetrahedral interstices among those three sites, forming a liquid behavior of Cu^+^ ions, which can be illustrated in Figure 1d, the ab initio molecular dynamics (AIMD) simulations. The liquid-like Cu^+^ ions suppress the propagation of heat transferred by phonons, shorten the phonon mean free path, and reduce the lattice thermal conductivity [30,35,37,38,39], while the ordered Se sublattice maintains the carrier transport, keeping the “electron crystal” characteristic. Thus, thermal conductivity is strongly depressed and electrical conductivity is intact, ensuring the high ZT value of Cu_2_Se bulk materials. However, even if it has a relatively high ZT value, the commercial application (ZT value higher than 3) is still far from reach. Hence, investigations on adjusting the TE performance of Cu_2_Se by doping has been conducted by many scientists. Moreover, practices proved that dopants do have significant effects on the TE performance of Cu_2_Se. Until now, the highest ZT value of 2.62 has been achieved in Al doped Cu_2_Se bulk materials at 1029 K [40], and it is found that the introduction of Al can shift the microstructure of Cu_2_Se, then influence the TE performance. Herein, various of dopants from all main groups and some transitional groups of elements that are used to dope Cu_2_Se are introduced. Fabrication methods, micro-/nano-structures, and their influences on Seebeck coefficient, electrical conductivity, thermal conductivity are presented. The doping effect and probable mechanisms on TE performance of Cu_2_Se are analyzed and discussed in detail. The main part of this article is organized and divided by groups of doping elements.

## 2. Doping by Group IA and IIA Elements

In year 2018, a high ZT value of 2.14 at 973 K was obtained in the Li doped Cu_1.98_Li_0.02_Se bulk materials through hydrothermal synthesis and hot-pressing by Qiujun Hu et al. [41]. The incorporation of Li^+^ ions, to some extent, influenced the crystal structure and microstructure of bulk Cu_2_Se. As is shown in Figure 2a, the incorporation of Li+ ions, with radius slightly smaller than the Cu+ ions, reduced the lattice parameter and cell volume of the matrix. And as we can see in the SEM image in Figure 2b, showing the microstructure of the bulk materials, nanopores came into being at the grain boundaries and then spread inward to the whole matrix with the increment of Li content. These changes in the crystal and micro structures explain the high ZT value of Cu_2_Se bulk materials in three ways: First, Li^+^ ions with radius smaller than Cu^+^ ions introduced huge amount of point defects with positive valence, which increase the strength of carrier scattering, i.e., increase the Seebeck coefficient; Second, the mass difference between Li^+^ ions and Cu^+^ ions engender the fluctuation of the mass field and stress field, whence, Li^+^ ions become the center of strong phonon scattering, which can lower the lattice thermal conductivity; Finally, nanopores introduced by Li dopants not only will enhance carrier scattering, but also will reduce the lattice thermal conductivity by lowering the sound velocity and phonon mean free path [42] (as can be derived by Equation (3)). At the same time, Li as an electron donor can annihilate the holes in Cu_2_Se matrix, causing a reduction in the hole concentration, then increasing the Seebeck coefficient. However, this action will also sacrifice the electrical conductivity. Figure 2c–f shows the thermoelectric experiment results of the samples, with which we can conclude that Li dopant drastically enhances the ZT value of the Cu_2_Se bulk material through decreasing thermal conductivity and increasing Seebeck coefficient.

Similar results are founded in Cu_2_Se materials with other alkali metal dopants. Zheng Zhu et al. [43,44] get the same nanopore microstructure in the Na and K doped Cu_2_Se bulk materials through the same synthesis approaches as Li doped samples. ZT values of 2.1 at 973 K and 1.19 at 773 K were achieved for Cu_1.96_Na_0.04_Se and Cu_1.97_K_0.03_Se respectively. The roles of Na and K played in the Cu_2_Se are similar to Li. On the one hand, the introduction of Na and K lowers the hole concentration in the Cu_2_Se matrix, then increases the Seebeck coefficient; on the other hand, Na^+^ and K^+^ point defects can enhance the scattering of phonons and carriers then decrease the thermal conductivity. What’s more important is, nanopores found both in Na and K doped Cu_2_Se materials can lower the sound velocity v and phonon mean free path l [42,45]. Moreover, it should be mentioned that Zheng Zhu et al., stated a new finding that the thermal capacity $C_{v}$ of their samples with Na, K dopant has an inversely proportional relationship with pore density, which is not pointed out in the Qiujun Hu’s work. Same experiments are carried out by Zhu Zheng’s group on copper selenide with Cu deficiency (Cu_2-x_Se) as well [46]. However, the ZT value of their sample Cu_1.94_Na_0.04_Se only reached 1.32 at 973 K, which is far less than stoichiometric Cu_2_Se doped by alkali metals. This result might be due to the loss of Cu^+^ ions decreases the strength of scattering of phonons, hence the thermal conductivity of Cu_1.98_Se is much higher compared to Cu_2_Se. In this case, even at the same level of doping content, ZT value of Cu_1.94_Na_0.04_Se is lower than Cu_1.96_Na_0.04_Se. Yingshi Jin et al. [47] also conduct experiments on Na doped Cu_2-x_Se, but with different synthesis method. The approach they used to consolidate the nano-powders into bulk materials is spark plasma sintering (SPS) other than hot-pressing. As a result, possibly due to the different synthesis approach, the final sample they acquired doesn’t possess the nanopores. Interestingly, the ZT value obtained in Jin’s sample only reach the maximum of 0.24 at 673 K. Regardless of the temperature difference, the divergence in the experimental outcomes between hot-pressing and SPS, to some extent, demonstrate the extraordinary role nanopores played in enhancing TE properties of copper selenides.

Alkaline-earth metals possess chemical and physical properties similar to alkali metals, such as small ion radius, highly reactive electron donors and etc. Empirically, we may deduce that alkaline-earth metals can perform as perfect as alkali metal dopants do in Cu_2_Se. Unfortunately, to our best of knowledge, few investigations on alkaline-earth metals doping on Cu_2_Se are conducted, and the only one we found using Mg as dopant doesn’t show the same enhancement as alkali metal dopants. But some interesting anomalies should be noted that Mg who should be an electron donor, on the contrary, increases the hole concentration in the Cu_2_Se matrix, and then reinforced the electrical conductivity, as displayed in Figure 3b,c. Moreover, Mg^2+^ ion with radius of 0.66 Å, a lot smaller than Li^+^ ions (0.76 Å) and Cu^+^ ions (0.77 Å), should introduce a huge amount of point defects, i.e., phonon scattering centers into the Cu_2_Se matrix, with which the thermal conductivity of Cu_2_Se is supposed to be undermined. Nevertheless, as shown in Figure 3d, the lattice thermal conductivity almost remains untouched and the carrier thermal conductivity even increased as a result of the higher carrier concentration. With reinforced electrical conductivity, Mg doped copper selenide reached a relatively high ZT value of 1.6 at 860 K, but at temperatures below 800 K, Mg doping shows a counteractive effect on the TE performance.

## 3. Doping by Group IIIA Elements

In 2014, a high ZT value of 2.62 at 1029 K Cu_2_Se bulk material with Al dopants was synthesized by Bin Zhong’s group [40]. The semiconductor crystal they acquired, as shown in the XRD pattern in Figure 4a, is made up of α-Cu_2_Se and Cu_3_Se_2_ composite at low temperature range and β-Cu_2_Se single phase at high temperature range. From Figure 4b the SEM combined with Figure 4c the EDS analysis, we can grab the information that this bulk material has the nature of highly aligned lamellae with the size of 3–4 μm, and all compositional elements has a homogeneous distribution along the lamellae and heterogeneous distribution across the lamellae. These peculiar microstructures lead to anisotropic TE properties of the samples. As is shown in Figure 4d, at the orientation across the lamellae, namely the direction normal to the aligned grain planes, the samples will always have a higher electrical conductivity and a lower thermal conductivity, both lead to a higher ZT value. The mechanism is not investigated or presented by Bin Zhong et al., but they stated that it stems from the highly aligned Cu^+^ ion layers in the aligned lamellae, which lower the lattice thermal conductivity. Similar results are found in Ting Zhang’s work [49]. They obtained the same microstructure of highly aligned Cu_2_Se material, but without any dopants. They proposed that it’s the aligned lamellae structure provide the Cu^+^ ions with more passageways for movements, making them capable of scattering more phonons and carriers. In this way, the lattice thermal conductivity and electrical conductivity will be decreased, compared to lamellae-structure-free samples. Also a Pb doped Cu_2_Se material synthesized by Zheng Zhu et al., has the same layered structure [50]. In their experiments, they find that the lower the width of the lamellae, the lower the thermal conductivity. Thus, they deduce that it’s the large amount of grain boundaries across the lamellae contribute to the remarkable decrease of the thermal conductivity. The phonons and carriers are efficiently scattered by grain boundaries, the thinner the lamellae, the denser the grain boundary, the lower the thermal conductivity. Both Ting’s and Zheng’s work explained, to some extent, the origin of the high ZT value. But no anisotropy is found in either of their works, which is boiled down to the randomly oriented grains, i.e., the lamellae in Ting’s or Zheng’s work are only locally aligned. However, it’s noteworthy that the large amount of grain boundaries across the lamellae should have strong scattering for both phonons and carriers, which will definitely lead to the decrease of both thermal and electrical conductivity. This is supported by result of Ting’s and Zheng’s work. However, in Bin Zhong’s work, an interesting anomaly is found that the electrical conductivity is higher on the direction perpendicular to the lamellae than along the lamellae, even with grain boundaries of high density.

Another dopant in group IIIA which have an extraordinary effect in enhancing the TE performance of Cu_2_Se materials is Indium. The Cu_2_Se/CuInSe_2_ (CS/CIS) nanocomposites, synthesized by A. A. Olvera et al., through ball milling and uniaxial hot-pressing, obtained the ZT value of 2.6 at 850 K for the sample Cu_2_Se doped with 1 mol% In, which is displayed in Figure 5i [51]. Different from other dopants, this time, the introduction of indium induces a formation of second phase in the Cu_2_Se matrix. As is shown in Figure 5a, the backscattered electron image of the sample with content of half CS and half CIS, a mechanical mixture is formed on a microscale. Areas with higher contrast, marked in blue, is CIS phase, and CS for lower contrast. Zooming in at CS areas, shown in Figure 5b,c, we can still find that the lower contrast region, which is used to be thought of CS region in Figure 5a, is still a mixture of CS and CIS on nanoscale. Moreover, interfaces of coherency are also found in the STEM image (Figure 5d), and illustrated in the schematics shown in Figure 5e. As for the TE properties, as is shown in Figure 5f,g, the introduction of the second phase CIS, plays an important role in enhancing the electrical conductivity and reducing the thermal conductivity. For the increase in the electrical conductivity, from the schematics in Figure 5e, the In^3+^ ions occupy the Cu^+^ ion sites, forming a hybrid “Cu_2_Se-like” substructure, play a part in confining the diffusion of Cu^+^ ions, give rise to an undercut in the scattering of carriers, which means a higher carrier mobility, also means a higher electrical conductivity. As for the reduction of thermal conductivity, the huge reduction, is a comprehensive outcome of the CIS nano-inclusions and coherent interfaces. Seen in Figure 5a,b, two phases of CIS and CS interweave from nanoscale to microscale, plenty of CIS nano-inclusions are formed in CS grains, becoming a perfect center of phonon scattering. At the same time, coherent interfaces formed between the nano-inclusions and CS matrix play an important part in prohibiting phonon transport on account of the different lattice vibration and the Cu^+^ disorder level between CS and CIS, but have a good performance in transferring carriers between those two phases. In this case, the TE properties is, almost idealized, improved by increasing the electrical conductivity and decreasing the thermal conductivity simultaneously.

Other elements in group IIIA are also used as dopants in Cu_2_Se, though with inferior ZT value to ones discussed above, interesting mechanisms that enhance the TE properties are quite worthy of mentioning. Cu_2_Se with nano-boron incorporated is carried out by Sheik Md. Kazi Nazrul Islam et al. [52], through ball-milling and SPS. Nano-boron particles are precipitated at the grain boundaries, as is shown in schematics, HAADF STEM image and EDS map in Figure 6a–e. Those insolating precipitants works as perfect phonon scattering centers, as a result, the thermal conductivity of the Cu_2_Se is significantly decreased, shown in Figure 6f. Even though the final highest ZT value of this sample only reached 1.6 at 852 K, much lower in comparison to Al and In doping, the ZT enhancement ratio is relatively high which is by a factor of 1.6 to 2.6 compared to the undoped sample. Thus, B is still a promising dopant for ZT enhancement of Cu_2_Se, if other synthesizing methods like hydrothermal methods and hot pressing are exploited to increase the ZT value of the original undoped sample, it is likely that the B doped samples can achieve better results. Ga doping is also investigated by F.S Liu et al., through ball milling and SPS [53]. A second phase CuGaSe_2_ is formed in the matrix, but Liu found this second phase have an opposite effect which is increasing the thermal conductivity while decreasing the electronic conductivity, and the increment of ZT value of their samples is not distinct, even it is decreased at some certain content.

## 4. Doping by Group IVA Elements

Comprehensive investigations on Cu_2_Se have been conducted by Lanlin Zhao et al. doped by a variety of carbon resources, including graphite (G), carbon nanotubes (CNT), super P (SP) and hard carbon (H) [54]. Among these, 0.3 wt% carbon fiber doped Cu_2_Se achieved the highest ZT value of 2.4 at 850 K. The composites are synthesized through solid state reaction and melt quenching. It is interesting to note that, from Figure 7a, the final composite synthesized by melt quenching, shows no dependence on the initial carbon sources, i.e., no matter what kind of carbon source is chosen, the eventual samples are composed of hexagonal carbon and monoclinic Cu_2_Se at room temperature. As is shown in Figure 7b,c, where the Cu_2_Se matrix are composed of small grains with sizes from 30–60 nm, and the carbon nanoparticles are gathered in grain boundaries. During the melt-quenching process, carbon precipitates played an important role in preventing grain ripening. As displayed in Figure 7d, Cu_2_Se with various carbon dopants shows an overall increase in the electrical conductivity and decrease in the thermal conductivity, leading to enhanced ZT values. The electrical conductivity increment can be apparently attributed to the high electrical conductivity nature of carbon. As for the thermal conductivity, which is dominant in enhancing the ZT values, is due to the huge amount of grain boundaries with carbon decorates. Those carbon nanoparticles, as indicated in the schematics in Figure 7e, can scatter phonons, resulting in a significant increase in the thermal boundary resistance, namely decrease in thermal conductivity. This mechanism is quite similar to boron-doped samples discussed in the section above.

Other experiments on C doping are conducted as well. Raghavendra Nunna et al. [55] synthesized a CNTs-Cu_2_Se composites through ball-milling and SPS. In their experiments, CNTs works similarly as the nucleation center, depressing the ripening of Cu_2_Se grains. However, different from the result in Zhao’s work, CNTs in Nunna’s experiments, maintained the original multiwalled-nanotube structure as presented in Figure 8a–e, even so, the mechanism in decreasing the thermal conductivity is identical by strong phonon scattering at CNTs enriched grain boundaries, which is displayed in Figure 8c. Finally, a high ZT value of 2.4 at 1000 K is obtained. The same mechanism is also supported by the work of Meng Li et al. by dispersing graphene nanoplates (GNPs) into the Cu_2_Se matrix through ball-milling and SPS [56]. Increased electrical conductivity and decreased thermal conductivity are brought about by the high density of grain boundaries with enriched GNPs. A high ZT value of 1.7 at 873 K is obtained. Meng Li et al. also investigated on the carbon nanodots (CDs) incorporated Cu_2_Se [57]. This might be due to the different synthesis methods which are hydrothermal synthesis and hot-pressing, the samples show a characteristic of layered structure. Apart from the phonon scattering at the interfaces between the CDs and Cu_2_Se matrix, those interfaces between the layers also play a part in reducing the thermal conductivity. Eventually a high ZT value of 1.98 at 973 K is achieved.

Apart from using expensive ingredients like carbon-nanotubes or graphene, novel approaches using grape juice as carbon resources are proposed by Sheik Md Kazi Nazrul Islam et al. [58], and an extraordinarily high ZT value of 2.5 at 984 K is obtained. In their synthesis process, Cu_2_Se powders are mixed with grape juice and dried up on a hot plate. Then the samples are calcined into bulks and heated up to 1200 °C. Finally, a polycrystalline Cu_2_Se structure, with C and Cu2O enriched boundaries is obtained, which is displayed in the STEM image in Figure 9a–d. Moreover, zooming in at the region of grain boundaries, Cu_2_O can be found at the interface between C and Cu_2_Se. Those Cu_2_O inclusions and carbon particles, together formed a carbonaceous material at grain boundaries, as illustrated in the schematics in Figure 9e. The TE performance of the samples are displayed in Figure 10. The existence of C and Cu_2_O, can increase the hole concentration and enhance electrical conductivity (seen in Figure 10a,b). In addition, the carbonaceous material can decrease the thermal conductivity by strengthen the phonon scattering at grain boundaries, which is illustrated in Figure 10d, the thermal conductivity measurement is displayed in Figure 10c. Thus, from Figure 10e, an overall enhancement of ZT value can be observed in all grape juice ‘doped’ samples, and a good thermal stability can be observed in Figure 10f.

Si is also used as dopants in adjusting the TE performance of Cu_2_Se. Jingdan Lei et al. introduced SiC nanoparticles into Cu_2_Se matrix, obtaining a remarkable ZT value of 2.0 at 875 K [59]. Their samples are synthesized through mechanical alloying and SPS. The final pellets, as presented in Figure 11a–c—the FESEM and BSE images of the sample, are comprised of nanograins of sizes from tens to hundreds of nanometers, and the ball shaped SiC nanoparticles are invisible in this image, as a result of the low mass of SiC compared to the Cu_2_Se matrix, but a ball-shape can be seen in the region marked by yellow circles, indicating the existence of SiC nanoparticles, which is also supported by the ball shaped Si in the EDS images, shown in Figure 11f. The TE properties are displayed in Figure 12a–d, the enhancement in the TE properties is attributed to mainly two aspects: the decrease in the thermal conductivities and the increase in the electrical conductivities, and these two aspects both benefit from the introduction of nano-SiC particles. These nano-particles brings about the production of more defects like nanotwins, facilitating the liquid behavior of Cu^+^ ions, i.e., those Cu^+^ ions–the ion carrier–are more prone to move out of the lattice thus increasing the electrical conductivity, as can be observed in Figure 12c. At the same time, the enhanced liquid behavior will contribute to a stronger phonon scattering. Plus, the dense grains and SiC nanoparticles are also perfect scattering centers for phonons of medium wavelength. Finally, the Cu_2_Se mixed with 0.05 wt% SiC nanoparticles achieves the highest ZT value of 2.0 at 875 K.

A Sn doped Cu_2_Se is synthesized through melting, annealing and SPS by Trevor P. Bailey et al., and an enhanced ZT value of 1 is observed in sample Cu_2(1−x)_Sn_x_Se at 823 K [60]. Sn^2−^ point defect and SnSe second phase both contribute to reducing thermal conductivity through strengthening phonon scattering. Though the TE performance of Sn doping is not as significant as C doping, Sn is one of the few elements discussed in this article that has a peculiar effect in chemically stabilizing the Cu_2_Se matrix at high temperatures and under a current by preventing Cu precipitation. The mechanism is: First, point defects formed by Sn atoms, which has a larger size than Cu atoms, play as physical and electrical barriers in hampering the migration of Cu atoms. Those point defects can cause distortion in the lattice blocking the pathway for Cu electromigration. Second, the SnSe second phase can also play a role in hindering Cu-ion transport. Chemical stability is one of the biggest barriers hindering Cu_2_Se from application, which will be talked about in the Analysis section. Similar study is also presented by F.S. Liu et al. [61]. In their study, through introducing SnSe into the Cu_2_Se matrix through vacuum melting and SPS, an enhanced ZT value of 1.41 at 823 K is obtained in (Cu_2_Se)_0.97_(SnSe)_0.03_ sample.

Pb is also used in doping of Cu_2_Se to enhance the TE property. Zheng Zhu et al. synthesized Pb-doped Cu_1.98−x_Pb_x/2_Se by hydrothermal synthesis and hot-pressing, obtaining a ZT value of 1.52 at 973 K [50]. The microstructures show a highly aligned layered structure similar to Al doped Cu_2_Se samples. They draw the conclusion that, by Pb doping, the carrier concentration is decreased, and more point defects and interfaces are introduced into the matrix.

## 5. Doping by VA Elements

Investigations on using VA elements as dopants are so few that the Bi is the only one we find. The Bi-doped α-Cu_2_Se (Cu_2-3x_Bi_x_Se) synthesized through hydrothermal method and SPS by Wangwei Liao et al., shows a ZT value of 0.43 at 373 K [62]. Bi as an electron donor, can reduce the carrier concentration of Cu_2_Se from $4.1\times 10^{20}$ to $2.0\times 10^{20}{\;\mathrm{cm}}^{-1}$ at a doping level of x = 0.006, but will increases the carrier concentration significantly if the Bi concentration is continually increased. This can be explained by the fact that at low Bi concentration, the incorporation of trivalent Bi^3+^ ions can compensate for the Cu deficiency, then decrease the carrier concentration, but at high Bi doping levels, the low solubility of Bi in the Cu_2_Se matrix can regenerate Cu deficiencies thus the carrier concentration is increased. Hence, due to the reduced carrier concentration, the carrier thermal conductivity is reduced significantly and finally resulted in an enhanced figure of merit ZT value of 0.43 at 373 K.

## 6. Doping by VIA Elements

For dopants in VIA and VIIA groups, as they have a relatively high electronegativity, elements tend to attain negative electrovalence and replace lattice sites of Se^2−^ other than Cu^+^ as dopants we discussed before. The most typical dopant used in Cu_2_Se tuning the TE properties is Sulphur. Kunpeng Zhao et al., presented a Cu_2_Se_1−x_S_x_ (x = 0.02–0.12) polycrystalline material obtaining a high ZT value of 2.0 at 1000 K [63]. The samples were first prepared through element melting and sample annealing, then the annealed ingot was grounded into powders and remade into bulks by SPS. The samples possess a nature of single phase and stacking layered polycrystallinity as displayed in the SEM image in Figure 13a. The introduction of S atoms into the lattice, strengthen the bonding energy between the Cu^+^ ions and S^2−^ ions, confining the Cu^+^ ions from moving out of the lattice. As a result, as presented in Figure 13b, the vacancy formation energy is increased significantly, which means that the formation of Cu^+^ vacancies are suppressed, thus the hole concentration is depressed, seen in Figure 13c. This phenomenon generally has three evident effects on the TE properties, which is shown in Figure 14a–d: the decrease of the electrical conductivity and thermal conductivity, plus the increase of Seebeck coefficient. Moreover, the introduction of S, not only alters the bonding energy, but also shortens the bond length, inducing a strain field fluctuation in the lattice, and it is found that the transverse sound velocity of S-doped samples is lower than undoped samples, thus the lattice thermal conductivity is also reduced. For these reasons, the ZT value of S-doped Cu_2_Se is significantly enhanced with a peak value of 2.0 at 1000 K among the testing range. Further investigations are also conducted by Kunpeng Zhao’s group on S-doped Cu_2_Se with higher S concentrations (x = 0.2–0.7) with the same synthesis methods [64]. At high S concentration range, the same mechanisms of TE property enhancement are found and a maximum ZT value of 1.65 is obtained at 1000 K which is lower than the samples with lower S concentration. Lisha Xue et al. put forward a quick synthesis of S-doped Cu_2_Se in just 30 min using high-pressure and high-temperature (HPHT) technology. The samples possess a structure of stacking layers, abundant pores and lattice defects, which all contribute to undermining the lattice thermal conductivity. However, only samples with x = 0.3 obtained ZT enhancement at temperature under 650 K. Other samples seem to vote for S has an inverse effect on the TE properties. Moreover, at temperature higher than 650 K, all samples have a uniform decrease in the ZT value. Similar experiments have been conducted on the copper selenide with Cu deficiency, which is Cu_1.98_Se precisely. Muhammad Umer Farooq et al. obtained Cu_1.98_Se with Cu_2_S nano-inclusions though S doping [65]. Simultaneous enhancement of Seebeck coefficient and decrease of thermal conductivity were observed, as a result of the Cu_2_S-inclusion-induced external strain and grain boundaries. An improved ZT value of 0.9 at 773 K was achieved. Lanling Zhao et al., synthesized S-doped Cu_1.98_Se with S concentration x = 0.02–0.16 [66], and a result of overall decline in ZT value was also observed at temperatures higher than 600 K, similar to Lisha Xue’s work on Cu_2_Se.

Widespread investigations are also conducted on Te-doped Cu_2_Se materials. Sajid Butt et al., carried out a Te-doping Cu_2_Se through ball-milling and SPS, a high ZT value of 1.9 at 873 K (mid-temperature range) is achieved with Te concentration of 10 mol% [67]. As shown in Figure 15a, the XANE spectra of Cu_2_Te and Te doped Cu_2_Se, distinctive A, B, and C peaks, which are the characteristics of Cu_2_Te can be observed in all Te doped samples, demonstrate the existence of a Cu_2_Te second phase, which is undetectable for XRD, as is shown in Figure 15b. This second phase (nanoclusters) can be described in the cartoon in Figure 15c, in a hexagonal pattern with Te replacing the original Se sites. By introducing Cu_2_Te nanoclusters, the carrier mobility is undercut by enhanced carrier scattering and carrier concentration is decreased. As a result, the electrical conductivity is hugely decreased and the Seebeck coefficient is increased, the overall effect is that the power factor almost remains unchanged, seen in Figure 15d–f. Thus, the improvement of the ZT value should be owing to the decreased thermal conductivity. Seen in Figure 15g–i, the decrease of thermal conductivity is also on the carrier’s part, as the lattice thermal conductivity is almost untouched and the carrier thermal conductivity is significantly decreased due to the decrease of electrical conductivity. With this ultralow thermal conductivity, the ZT value is conspicuously improved in mid-temperature range as displayed in Figure 15j.

Kunpeng Zhao et al., also obtained the Cu_2_Te nanoclusters by Te doping with higher Te concentration (higher than 20 mol%) [68]. This time, the second phase can be detectable in the white dashed circle in SXRD image shown in Figure 16. Though the same evident decrease of electrical conductivity and thermal conductivity is observed, the overall ZT value is reduced compared to undoped sample. This might be due to the intrinsic higher electrical conductivity nature of Cu_2_Te, whence the increased concentration of Te, cut down the extent of reduction of the electrical conductivity and thermal conductivity, leading to a whittled TE performance [67]. At lower Te concentration(x = 0.01–0.10), regardless of the different synthesis method, the samples show no sign of the Cu_2_Te second phase [69,70]. Though the ZT value of the doped samples are enhanced, it is found that the enhancement is mainly due to the decrease of the lattice thermal conductivity caused by point defects and microstructures.

Interesting experiments of equally proportioned ternary copper chalcogenide composite was also carried out by Kunpeng Zhao et al. [71]. The final sample with a content of Cu_2−x_S_1/3_Se_1/3_Te_1/3_ (x = 0–0.03) shows a uniform element distribution and mosaic microstructure, as displayed in Figure 17a–c. The whole bulk was comprised of nanograins with small tilting angles to each other and sizes from 10 to 30 nanometers. Other than the thermal decline caused by point defects or lattice strains, this mosaic micro/nano-structure contribute to huge amount of grain boundaries, which contributes to further increment of thermal resistivity. Meanwhile, the electrical transport maintained intact due to the homogeneous distribution and small tilting angle between those mosaic nanograins. At last, a maximum ZT value of 1.9 at 1000 K is achieved when x = 0.02.

## 7. Doping by VIIA Group Elements

Doping by halogen elements seems to be less promising, as halogen ions usually possess an electrovalence of −1, which means a Cu^+^ vacancy should be formed when one halogen atom is doped into the lattice replacing the site of a Se^2−^ ion. While the deficiency of Cu^+^ has a significant reverse effect on the TE performance of Cu_2_Se [72,73] by increasing the thermal conductivity and reducing the Seebeck coefficient. Nevertheless, some achievements are still made in halogen doped samples, and some conflicts are quite interesting to be noted. Jingyi Wang et al. synthesized a series of Cu_2_Se_1−x_I_x_ through hydrothermal method and hot-pressing, obtaining a ZT value of 1.1 at 773 K, as shown in Figure 18d [74]. I^−^ ions and Cu^+^ vacancies introduced plenty of point defects into the lattice, resulting in an overall result of reduced thermal conductivity and enhanced electrical conductivity, as displayed in Figure 18a–c. While the case is different in Br and Cl doped samples. Though Cu deficiency can attribute to the increase of hole concentration [73], a significant decrease of hole concentration is observed in work of Tristan W. Day’s group on Br-doped Cu_2_Se due to the donor nature of halogen elements, resulting in a decrease of electrical conductivity [75]. The same decrease of electrical conductivity is also observed in work of Minji Kim et al. on Cl-doping [76]. Moreover, a second phase of CuCl is observed in Cl doped samples, which played a significant role in reducing thermal conductivity by strengthening phonon scattering. With the existence of the second phase, a high ZT value of 0.6 is achieved at 620 K, twice of the undoped sample. However, no second phase is observed in Br or I doped samples.

The second phase finding in Cl doped sample can be a promising approach to compensate for the negative effects brought by the unavoidable Cu deficiency. Nevertheless, an unexpected thermal instability is found at temperatures higher than 620 K, due to the low melting point of CuCl [76]. Even so, another discovery made by Huilin Liu et al. might be a way out [77]. In their study, Cu_2_Se was synthesized through SPS. A high ZT value of 2.3 is achieved at 400 K (shown in Figure 19) due to the phonon critical scattering during phase transition of materials possessing the nature of second-order phase transition. In a study of Br doped Cu_2_Se, Tristan W. Day et al. put forward that if the hole concentration in Cu_2_Se can be decreased to an appropriate concentration, the ZT value of Cu_2_Se can be increased to as high as 1.16 at 305 K theoretically [75]. Thus, halogen doped samples can still possible be made in near-room-temperature TE materials.

## 8. Doping by Transition Metals

Transition metals also play an important part in Cu_2_Se TE performance enhancement, and might be due to the similar properties of Ag to Cu, many investigations are conducted on Ag doped Cu_2_Se. While divergences are observed in the influence on TE performance of Cu_2_Se by Ag doping. The Cu_2−x_Ag_x_Se (x = 0.01, 0.02) synthesized by Weidi Liu et al. through solvothermal method followed by SPS shows a reduction in the ZT value in the whole temperature range [78]. In their study, they observed that the introduction of Ag can increase the hole concentration and decrease the hole mobility, in which the latter is predominant resulting in a reduction in the electrical conductivity. What’s more important is, the substitution of Cu^+^ with Ag+ leads to an enhancement in the thermal conductivity. This can be explained by the schematics in Figure 20a, since Ag+ has a larger ion radius and mass, it is more difficult for Ag^+^ to diffuse in the lattice, as a result, those point defects formed by Ag+ ions can hamper the diffusion of Cu^+^ ions, which undercut the liquid-like behavior of Cu^+^ ions, namely the strength of the phonon scattering induced by Cu^+^ ions is weakened. For these reasons, a reduced ZT value were observed in Ag doped samples. The same reduced ZT value is observed in work presented by Sedat Ballikaya et al., at high temperatures with higher concentration of Ag (x = 0.2) [79]. The samples are synthesized through melting and annealing plus SPS. However, in Sedat’s samples, an overall reduction of lattice thermal conductivities at all temperatures is observed, which they ascribe to the phonon scattering by Ag induced point defects, interestingly contrary to the conclusion drawn by Weidi Liu’s group. The ZT curve presented by Sedat’s group (Figure 20b) shows a ZT enhancement at temperatures below 700 K, and maximize at 650 K of 0.52, the suppression of ZT value at temperatures above 700 K is owing to the intrinsic conduction of the sample and depressed electrical conductivity. Enhanced TE performances are also obtained through Ag doping. The Cu_1.86_Ag_0.14_Se synthesized through a solvothermal method and SPS, by Weidi Liu et al., acquires an improved ZT value of 1.1 at 773 K [80]. This figure of merit was achieved mainly due to the extremely low lattice thermal conductivity $\kappa_{l}$ of 0.23W/m·K produced by the high density of pores that formed in SPS process, shown in Figure 20c,d. There are other types of Ag doped samples that also showed enhanced ZT values. For instance, A.J Hong et al., obtained a CuAgSe single phase crystal, which can be regarded as Ag doped Cu_2_Se at 50 mol% doping level, that reaches a high ZT value of 0.95 at 623 K [81]. Mengjia Guan et al., doped Cu_2_Se with Ag and S at the same time, achieving an improved ZT value of 1.6 at 900 K. Both of these enhancements are the result of the reduced thermal conductivity.

Other efforts have also been made on transitional elements doped Cu_2_Se. Erying Li’s group did investigations on the Hg doped Cu_2_Se by hydrothermal synthesis and hot-pressing [82]. A maximum ZT value of 1.5 is achieved at 773 K. By doping Hg, which have a different mass and electrovalence compared with Cu, increases the carrier scattering and brings more Cu^+^ deficiency into the lattice, thus the carrier mobility is decreased and the carrier concentration is increased. However, the latter is predominant, hence an increased electrical conductivity is observed. Moreover, the lattice thermal conductivity is also decreased owing to the point-defect- and dislocation-induced phonon scattering enhancement. An overall improvement in ZT value is observed in all Hg doped samples within the whole temperature range. A similar synthesis process on Ni doped Cu_2_Se is also produced by Feng Gao et al. [83]. of the same group. A maximum ZT value of 1.52 is obtained at 873 K. This enhancement is mainly due to the lattice thermal conductivity decrease induced by the formation of the Ni_0.85_Se second phase. P. Peng and his co-workers did a full research on doping effect of Cu_2_Se by transitional elements, which are Fe, Ni, Mn, Zn, and Sm, through a synthesis process of melting, ball-milling plus quenching, and samples are made into bulks through SPS [84]. In their experiment, all doped samples show a ZT enhancement and the highest ZT value of 1.07, 1.51, 1.28, 1.25, and 1.07 is observed at 823 K. TE performance enhancement by Zn and Ni doping is also supported by other groups’ works [85,86].

## 9. Analysis

Best TE performances achieved through doping by elements discussed above are summarized in Table 1. From discussions all above, in order to impart Cu_2_Se with higher TE performance, some characteristics are of necessity for any kinds of dopants: (i) the enhancement of Seebeck coefficient; (ii) increment of the electrical conductivity; (iii) the reduction of the thermal conductivity. Typically, due to the intrinsic “phonon-liquid” characteristic of Cu_2_Se, and the inverse relation between the electrical conductivity and the Seebeck coefficient, significant increase in the TE performance of Cu_2_Se is generally the main result of the reduced thermal conductivity κ, lattice thermal conductivity $\kappa_{l}$ in most cases. From all the examples shown above, reduction of thermal conductivity is brought about from 2 aspects: First, the peculiar microstructures, like the nanopores in Li doped samples, highly aligned lamellae structure in Al doped samples, mosaic composites structures in S, Te doped samples and etc.; Second, ubiquitous defects like inclusions, dislocations, low angle grain boundaries, deficiencies and so on. These microstructures and defects forms huge amount of scattering centers for both phonons and carriers, lowering the thermal conductivity conspicuously. It is absolutely of no doubt that dopants play an important role in TE performance enhancement of Cu_2_Se, and this has been discussed and investigated thoroughly by researchers. However, there are still some points that are worth noting if we want to take the TE performance of Cu_2_Se one step further. Firstly, the mechanism that a specific microstructure affects the TE performance, and the manipulation of microstructures. In the study of Lei Yang et al., bulk Cu_2_Se samples with stacked plate-like nanograins are produced through solvothermal method and SPS [87]. As they proposed that it’s the highly dense small-angle grain boundaries with dislocations that enhance the phonon scattering that produce an increased ZT value of 1.82 at 850 K. Bhasker Gahtori et al., also put forward that small grain sizes within nano- to meso-scale and high density of grain boundaries huge contribution to the enhancement of TE performance [88]. However, in the comparison experiment carried out by Lanlin Zhao et al., samples with different grain sizes and grain boundary densities are produced through melt-quenching [89]. By comparison, they find that samples with small grain sizes and high density of grain boundaries show no lower thermal conductivity than samples with larger grain sizes and low density of grain boundaries. They draw a conclusion that the TE performance of Cu_2_Se has low dependence on the grain size and the density of grain boundary, the high TE performance almost all comes from the liquid like behavior of Cu^+^ ions. This point of view is validified, to some degree, by the work of Ting Zhang et al. [49]. In their study, samples synthesized by powders with larger grain sizes possess higher ZT values. They conclude that it’s the aligned lamellae structure that provides Cu^+^ ions with more passage ways for movement, which benefits the scattering of both phonons and carriers, i.e., lead to a higher TE performance. Secondly, taking advantage of the second order phase transition from α to β phase may be a promising way of obtaining materials with high ZT value at near room temperature. The work presented in Section 6, the I doped Cu_2_Se obtained an ultrahigh ZT value of 2.3 at 400 K [77], which generally speaking, is obtained at temperatures around 1000 K for Cu_2_Se with enhanced TE performance. However, by making use of the phase transition nature of Cu_2_Se, high figure of merit can be obtained at 400 K. Apart from I doped Cu_2_Se, a huge ZT value exceeding 400 is achieved on undoped Cu_2_Se at phase transition temperature by Dogyun Byeon et al. [90]. And they gives more detailed explanation that this high ZT value originate from the huge Seebeck coefficient generated by the self-tuning carrier concentration effect. This effect, as we can anticipate, is related to the coexistence of α and β phase at phase transition temperature. Thirdly, the avoidance of Cu deficiency in the process of synthesis. The absence of stoichiometric Cu_2−x_Se with Cu deficiency are also investigated by many researchers, however a reverse effect on the TE performance is observed [72,73]. The introduction of Cu deficiency has little effect on the band structure and morphology of the sample, but can reduce the phonon scattering due to the reduced amount of Cu^+^ ions. Besides, as predicted in Kunpen Zhao’s work, the ZT value can be increased by decreasing the hole concentration, which is unfortunately increased with the introduction of Cu vacancies [63]. Though enhanced TE performance is observed in Cu_2_Se with Cu deficiencies in Triti Tyagi’s work [91], this enhancement stems from other factors like the nanoscale defects and abundant grain boundaries. What should be pointed out is that, in reality, no stoichiometry Cu_2_Se with Cu: Se ratio exactly at 2:1, Cu deficiencies is inevitable, thus, adjusting synthesis method which can reduce the Cu deficiency as much as possible can have positive influence on the enhancement of TE performance. In addition, there is doping of low-dimensional Cu_2_Se materials. In 1993, theoretical calculations proposed by L.D. Hicks et al., proved that low dimensional materials have better TE performance than bulk materials with the same composition [92,93]. And low-dimensional Cu_2_Se materials are also synthesized: the Poly(3,4ethylenedioxythiophene):poly(styrenesulfonate) coated Cu_2_Se (PEDOT:PSS/Cu_2_Se) nanowires produced by Yao Lu et al. shows a high ZT value of about 0.3 at room temperature[94]; the Cu_2_Se thin film synthesized through pulsed hybridreactive magnetron sputtering (PHRMS) reaches a high ZT value of 0.4 at room temperature [95]. Further doping these low-dimensional Cu_2_Se materials might improve the TE performance one step further. Besides, the chemical stability of Cu_2_Se [96]. The 3M corporation, General Atomics corporation, and NASA Jet Propulsion Laboratory (JPL) have done full research on the chemical stability of Ag doped Cu_2_Se. Three major problems are reported: First, the weight loss caused by Se evaporation; Second, precipitation of Cu and high reactivity at elevated temperature and under a current, for example, Cu in Cu_1.97_Ag_0.03_Se_1+y_ will precipitate on the surface and react with Fe if it’s in contact with stainless steel; Third, physical degradation caused by the electromigration under a current. Those problems will either have a inverse effect on the TE performance of Cu_2_Se or make it less durable. Appropriate solutions for those problems is as important as enhance its ZT value [97,98]. What’s more, other factors like different synthesis methods are also worthy of discussion. Cu_2_Se fabricated through different synthesis methods can possess a ZT value with enormous difference, due to the difference in nano- or micro-structures [91,99,100,101,102,103,104,105]. Choosing a proper synthesis method may have an unexpected effect on the final TE performance of Cu_2_Se.

## 10. Conclusions

To sum up, Cu_2_Se has an intrinsic high TE performance. Through doping, many elements, like Al, Li, Na, and In, etc., have made the ZT value of Cu_2_Se exceed 2, and almost break through the minimum requirement of commercial use (ZT ≥ 3). It is found that the high TE performance mainly originates from the fact that dopants can form point defects and dislocations, bring the mass fluctuation and strain fluctuation into the lattice, shift the microstructures, and so on, all of which play an important role in significantly scattering phonons and carriers (holes for Cu_2_Se specifically), reducing the thermal conductivity below the glassy limit. For future investigations and applications, from our point of view, exploring new dopants that can produce higher TE performance and optimizing existing ones are of equal importance. Further, we suggest that, when doping Cu_2_Se, researchers can pay attention to the manipulation of microstructures, the anomaly at phase transition, and the avoidance of Cu deficiency, with which higher ZT values may be produced.

## Figures and Tables

**Figure 1 materials-13-05704-f001:**
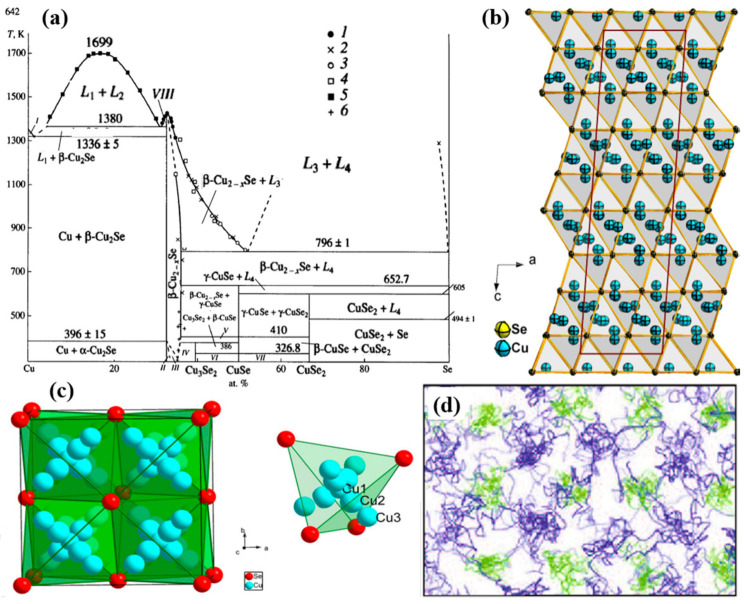
(**a**) T-x phase diagram of Cu-Se system, in which (I) Cu + $L_{1}$, (II ) α -Cu_2-x_Se, (III ) α -Cu_2-x_Se + β -Cu_2-x_Se, ( IV ) β -Cu_2-x_Se + Cu_3_Se_2_, (V ) β -Cu_2-x_Se + β -CuSe, (VI ) Cu_3_Se_2_ + α -CuSe, (VII ) α -CuSe + CuSe_2_, (VIII ) $L_{2}$ + β -Cu_2_Se [32]; (**b**) Crystal structure of α− Cu_2_Se with space group of C2/c, lattice parameters are a = 7.1379 Å, b = 12.3823 Å, c = 27.3904 Å, β = 94.3018°, blue balls represent Cu^+^ ions and yellow balls represent Se^2−^ ions [34]; (**c**) Crystal structure (unit cell) of β -Cu_2_Se, in which Se^2−^ are stacked in close-packed arrangement in < 111> direction, with layers in the sequence of ABCABC…, Cu1, Cu2 and Cu3 are three possible position of Cu^+^ ions with total occupancy of 1 [34]; (**d**) Atomic trajectories of Cu^+^ and Se^2−^ ions in β -Cu_2_Se in AIMD simulations for 3 ps at 700 K [37].

**Figure 2 materials-13-05704-f002:**
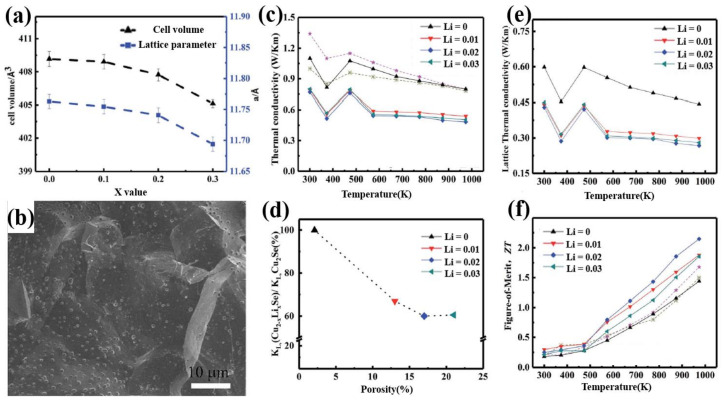
(**a**) Lattice parameters and cell volume of Cu_2−x_Li_x_Se (x = 0.1–0.3), a monotonous decrease can be observed when increase x; (**b**) SEM image of Cu_1.97_Li_0.03_Se; (**c**) Thermal conductivity of Cu_2-x_Li_x_Se with temperatures range from 300 K to 1000 K; (**d**) The relationship between lattice thermal conductivity $\kappa_{L}$ and porosity; (**e**) Lattice thermal conductivity of Cu_2−x_Li_x_Se with temperatures range from 300 K to 1000 K; (**f**) Figure of merit (ZT) of Cu_2-x_Li_x_Se with temperatures range from 300 K to 1000 K [41].

**Figure 3 materials-13-05704-f003:**
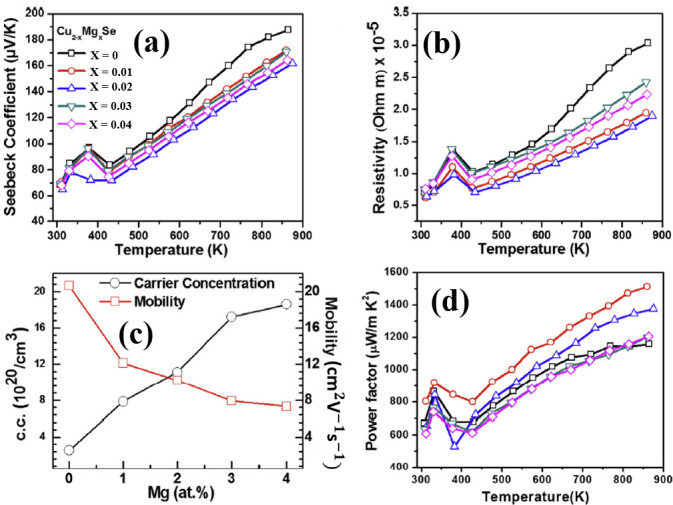
TE properties of Cu_2−x_Mg_x_Se (**a**) Temperature dependence of Seebeck coefficient; (**b**) Temperature dependence of resistivity $\rho =\frac{1}{\sigma }$ (σ is electrical conductivity); (**c**) Mg atom concentration dependence of carrier (hole) concentration and carrier mobility; (**d**) Temperature dependence of thermal conductivity [48].

**Figure 4 materials-13-05704-f004:**
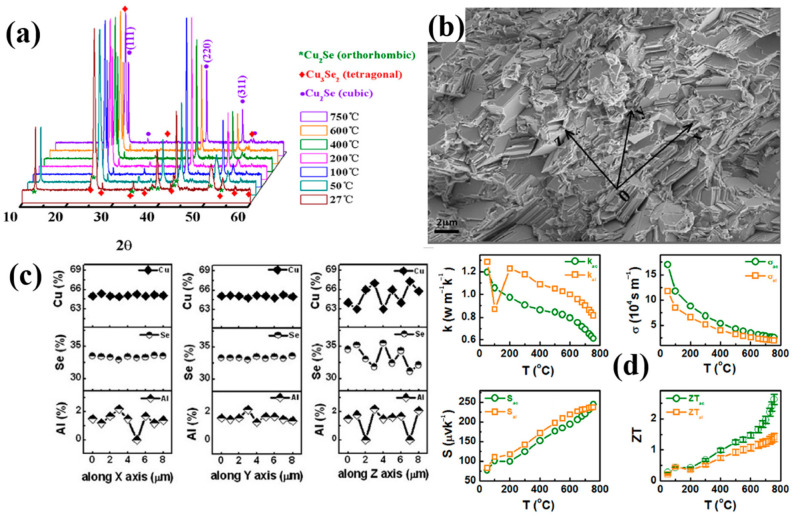
(**a**) XRD patterns of the samples at temperatures from 27 °C to 750 °C; (**b**) SEM image of the highly aligned lamellae structure at cross section; (**c**) EDS measurement result of the stoichiometry at X, Y, Z directions in (**b**), X, Y directions are along the lamellae and Z direction is across the lamellae; (**d**) Temperature dependence of TE properties across (ac) the lamellae, (shown in green legend) and along (al) the lamellae, (shown in orange legend); and the four figures are: upper left: thermal conductivity, upper right: electrical conductivity, down left: Seebeck coefficient, down right: figure of merit (ZT) [40].

**Figure 5 materials-13-05704-f005:**
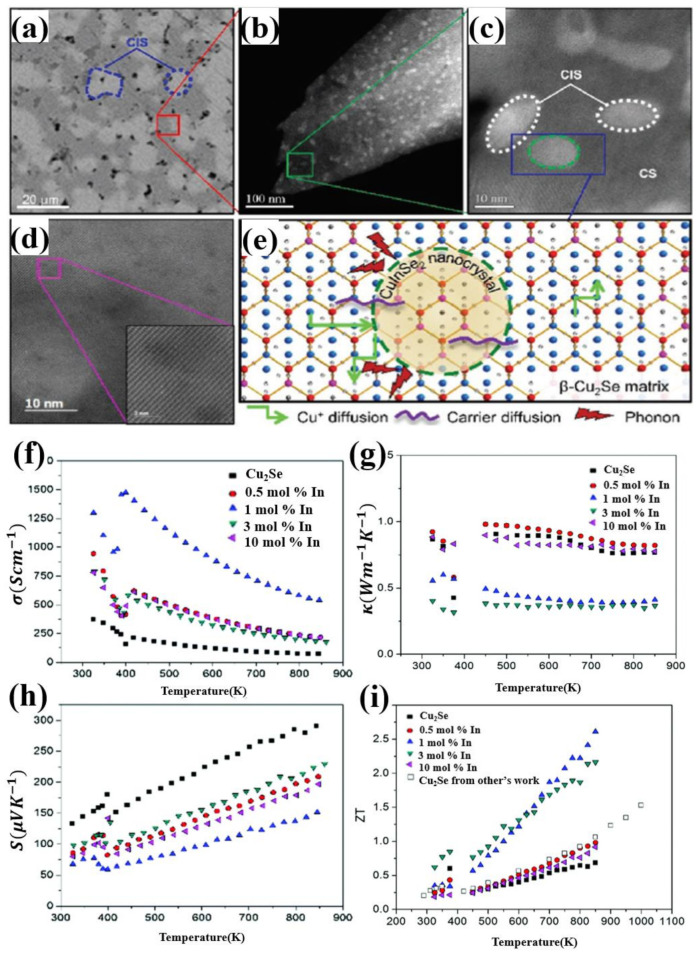
(**a**) Back scattered electron (BSE) image of the sample composed of 50% Cu_2_Se(CS) and 50% CuInSe_2_ (CIS), showing an interweaving structure of CIS and CS on a microscale; (**b**) HAADF-STEM image of the CS region enclosed in red in (**a**), showing a interweaving structure of CIS and CS on a nanoscale; particle sizes are under 10 nm; (**c**) higher magnification image of the region marked in green in (**b**), two distinct phase of CS and CIS can be observed; (**d**) STEM image of the sample with 1 mol% In, the inset shows a well-ordered atomic structure; (**e**) Schematics showing the lattice of CS and CIS composite and their interfaces, mechanism of phonon scattering, carrier diffusion as well as the Cu^+^ diffusion; (**f**) Temperature dependence of electrical conductivity; (**g**) Temperature dependence of thermal conductivity; (**h**) Temperature dependence of Seebeck coefficient; (**i**) Temperature dependence of figure of merit (ZT) [51].

**Figure 6 materials-13-05704-f006:**
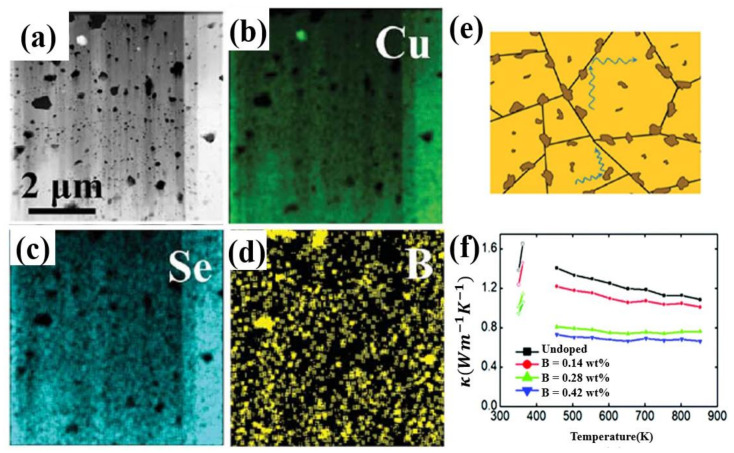
(**a**) HAADF-STEM image of 0.42wt% B doped samples, showing that black spots are boron-rich precipitates, gray background is Cu-rich matrix; (**b**–**d**) EDS mapping for (**a**); (**e**) Schematics of phonon scattering by B-riched nanoparticles at the grain boundary; (**f**) temperature dependence of thermal conductivity, in the inset the unit for B concentration is wt% [52].

**Figure 7 materials-13-05704-f007:**
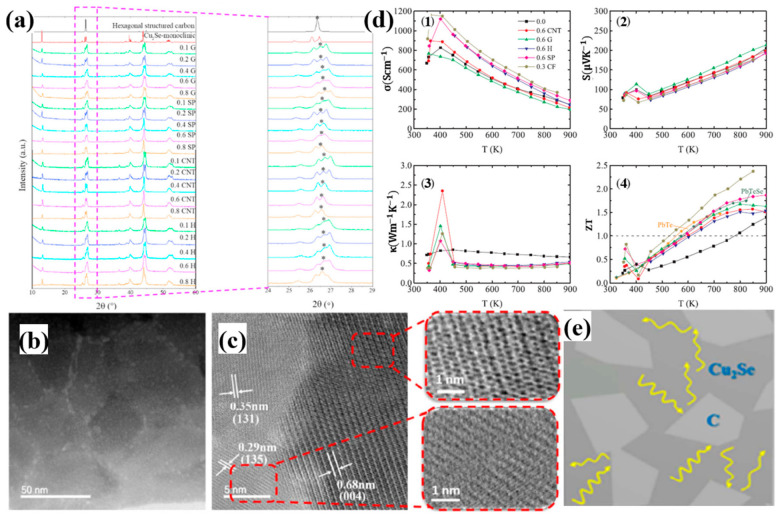
(**a**) XRD pattern for carbon doped Cu_2_Se with various kinds of carbon resources, which are graphite (G), super P (SP), carbon nanotubes (CNTs) and hard carbon (H); (**b**) TEM image with low magnification for 0.8wt% graphite doped Cu_2_Se, indicating that carbon nanoparticles are enriched at grain boundaries; (**c**) TEM image with high magnification, showing the lattice structure and interface between Cu_2_Se matrix and carbon nanoparticles, the honeycomb structure of graphite can be easily observed in the upper right inset of (**c**); (**d**) TE properties of doped samples: (1): electrical conductivity, (2): Seebeck coefficient, (3): thermal conductivity and (4) figure of merit (ZT); (**e**) Schematics of the microstructure of the doped samples and the process of phonon scattering [54].

**Figure 8 materials-13-05704-f008:**
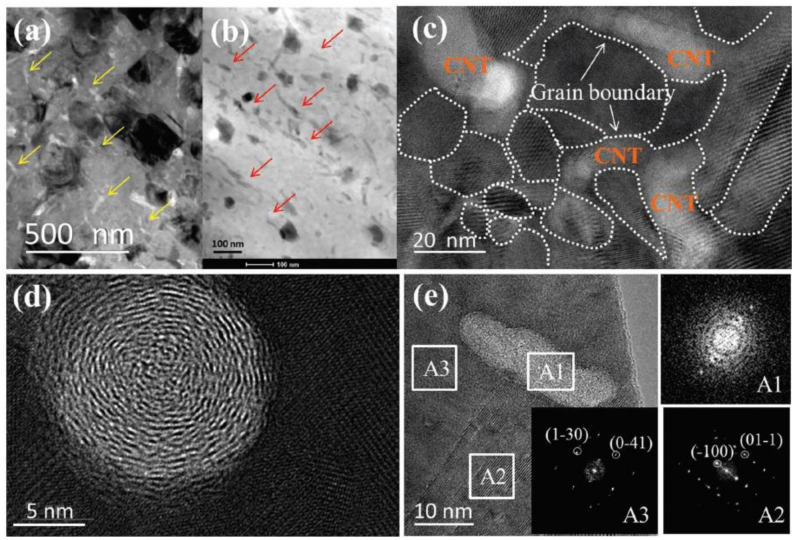
Structure characterization for samples with 0.75 wt% CNTs. (**a**) TEM image of CNTs doped samples with low magnification, yellow arrows are indicating carbon nanotubes with light contrast; (**b**) High angular annular dark field (HAADF) image, carbon nanotubes indicated by red arrows are in dark contrasts; (**c**) High magnification image, grain boundaries are denoted by white dashed lines; (**d**,**e**) observation of nanotubes along and normal to the tube directions respectively; insets A1, A2, A3 are Fast Fourier Transformation (FFT) patterns of regions marked in (**c**) [55].

**Figure 9 materials-13-05704-f009:**
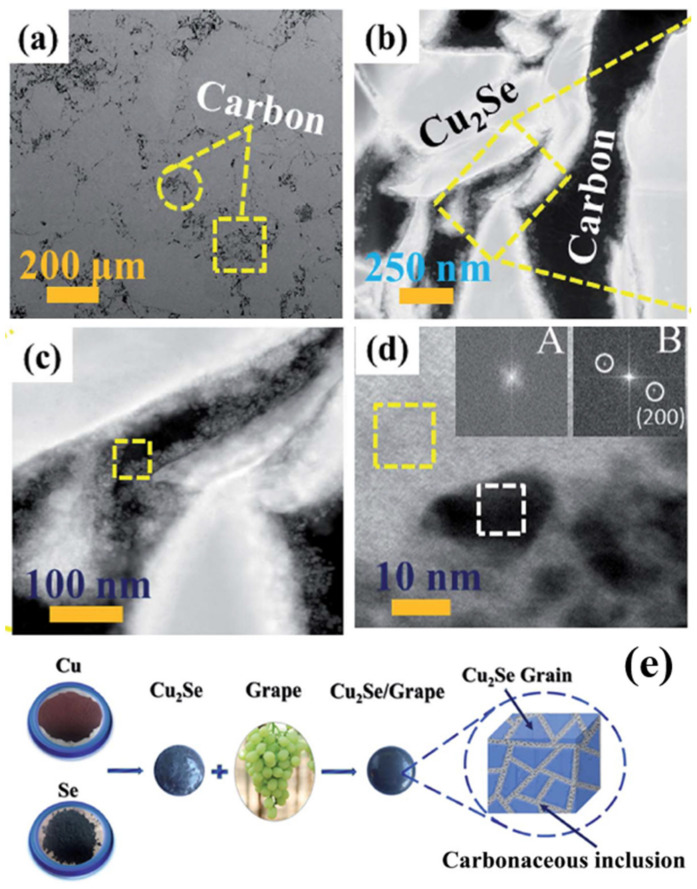
Structure characterization of sample doped with 0.6wt% C. (**a**) Secondary electron SEM (SE-SEM) image, presenting the grain boundaries and carbon phase, as can be seen, in which all carbon phases are located at grain boundaries; (**b**) HAADF-STEM image of the grain boundaries, and the carbon phase; (**c**) Magnified image of the yellow-dashed-rectangle-enclosed area in (**b**), the yellow dashed square enclosed region is indicating the Cu_2_O nanocluster inlaid in carbon phase; (**d**) Magnified image of the yellow dashed square in (**c**), region marked in yellow dashed square is amorphous carbon phase supported by inset A, region marked in white dashed square is Cu_2_O nanoparticles proved by inset B. Inset A and B are FFT image of those dashed squares, indicating the lattice diffraction patterns of marked areas; (**e**) the schematics of the synthesis process and the product; [58].

**Figure 10 materials-13-05704-f010:**
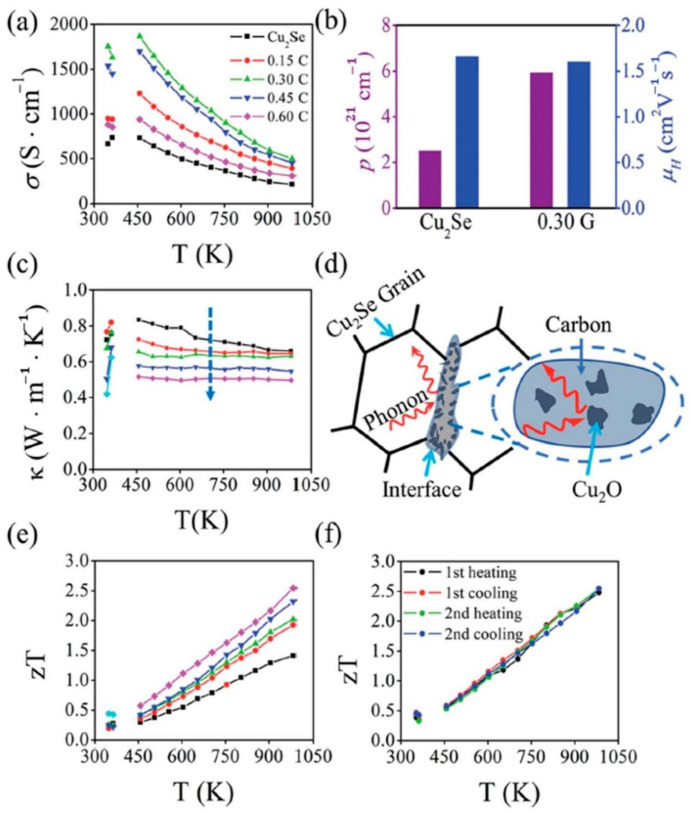
TE performance of all samples. (**a**) Temperature dependance of electrical conductivity σ; (**b**) carrier concentration (p ) and carrier (hole) mobility ($\mu_{H}$ ) of pure Cu_2_Se and 0.3 wt% doped Cu_2_Se (0.3 G); (**c**) Temperature dependence of thermal conductivity; (**d**) Schematics of phonon scattering by carbonaceous grain boundaries; (**e**) Temperature dependence of figure of merit (ZT); (**f**) Four repeated measurement of ZT value for 0.6 wt% C doped sample [58].

**Figure 11 materials-13-05704-f011:**
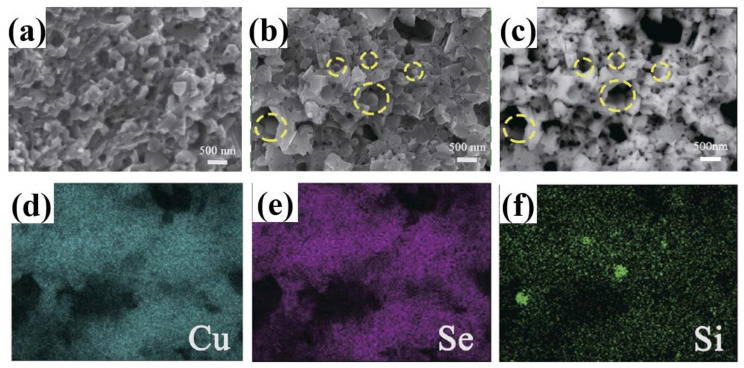
(**a**) Field Emission SEM (FESEM) image of Cu_2_Se; (**b**) FESEM image of 0.2 wt% SiC doped Cu_2_Se, regions enclosed by yellow dashed circles are indicating SiC particles which is invisible in FESEM image; (**c**) BSE image of (**b**), the SiC particles are still invisible due to the low mass of Si or C compared to Cu and Se; (**d**–**f**) EDS mapping of Cu, Se and Si, of the same area in (**b**), green spots in (**f**) proved the existence of SiC particles at yellow dashed circles in (**b**) [59].

**Figure 12 materials-13-05704-f012:**
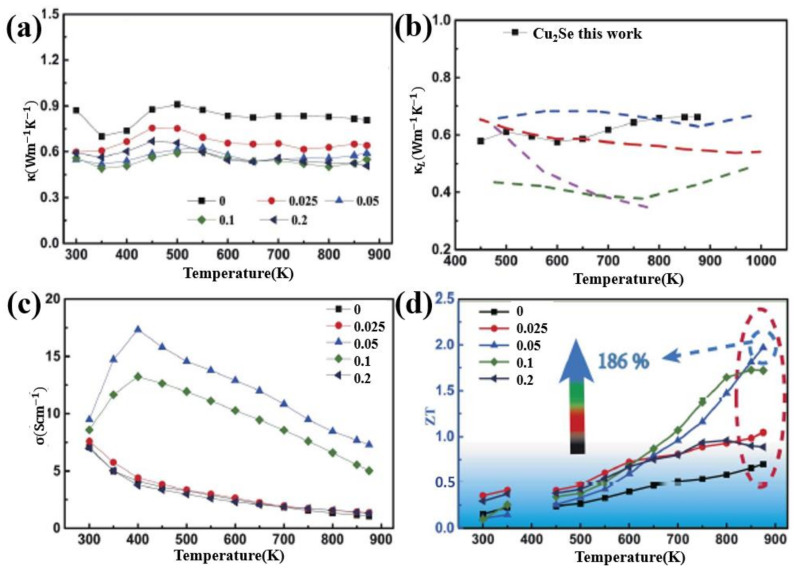
Temperature dependence of TE properties for Cu_2_Se/x wt% SiC (x = 0–0.2). (**a**) Thermal conductivity; (**b**) Lattice thermal conductivity; (**c**) electrical conductivity; (**d**) Figure of merit (ZT). Dashed lines are results of previous work on pure Cu2Se, which is not included in this review [59].

**Figure 13 materials-13-05704-f013:**
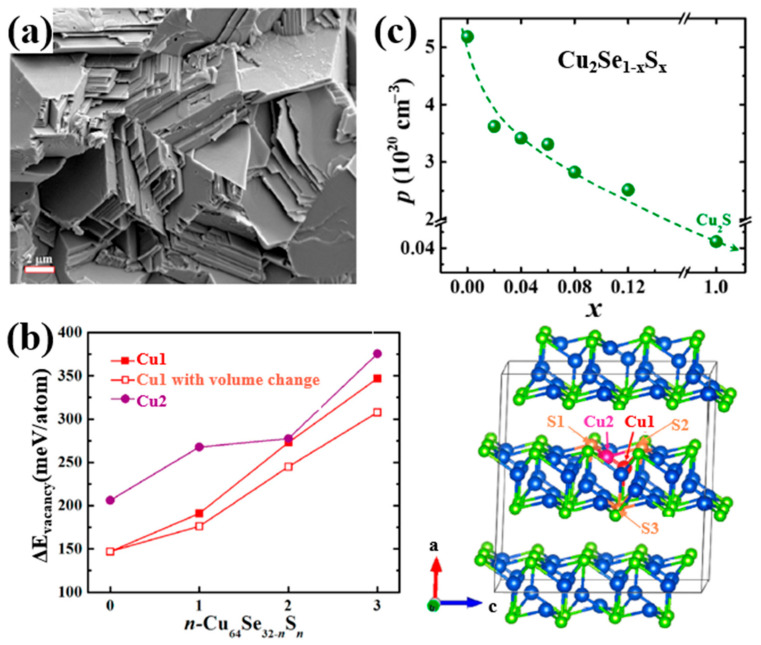
(**a**) SEM image of Cu_2_Se_0.88_S_0.12_, showing a stacked plates microstructure; (**b**) S concentration dependence of Cu vacancy formation energy ($\Delta E_{vacancy}$), Cu1, Cu2 are indicated in the lattice schematics on the right-hand side; (**c**) S concentration dependence of hole concentration (p ).

**Figure 14 materials-13-05704-f014:**
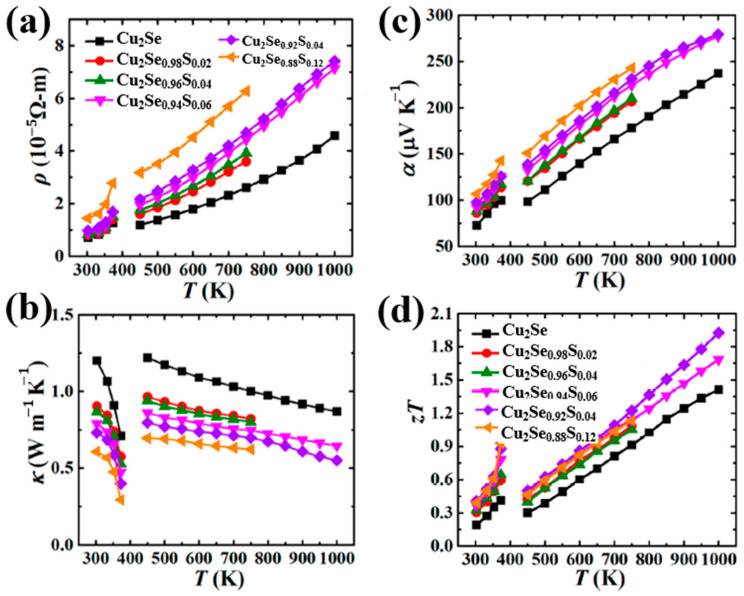
Temperature dependence of TE properties for Cu_2_Se_1−x_S_x_ (x = 0.02–0.12). (**a**) resistivity: ρ = 1/σ; (**b**) thermal conductivity κ (**c**) Seebeck coefficient α (the same with S in previous paragraphs); (**d**) Figure of merit (ZT) [63].

**Figure 15 materials-13-05704-f015:**
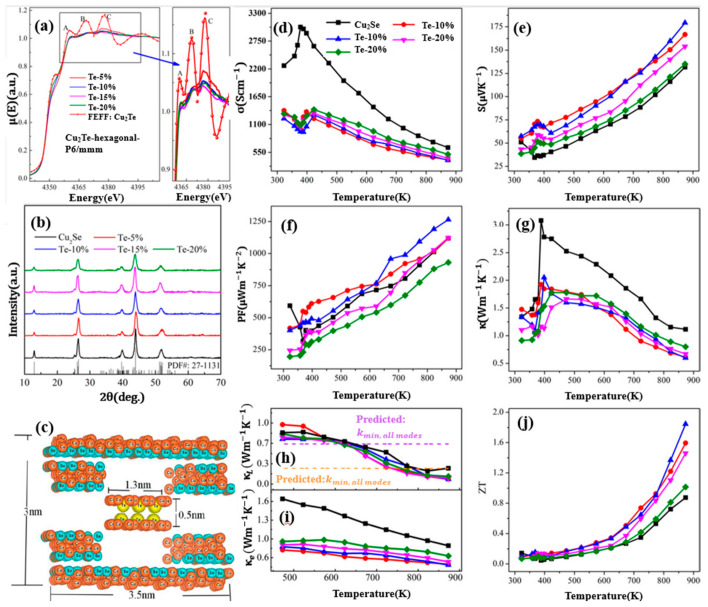
(**a**) XANE spectrum for pure Cu_2_Te and Te doped Cu_2_Se, characteristic peaks A, B, C for pure Cu_2_Te can be observed in all doped samples, which is shown in the magnified picture on the right; (**b**) XRD patterns for pure Cu_2_Se and doped samples, only single phases can be found; (**c**) Schematics indicating the Cu_2_Te second phase; (**d**) Temperature dependence of electrical conductivity; (**e**) Temperature dependence of Seebeck coefficient; (**f**) Temperature dependence of power factor (PF); (**g**) Temperature dependence of thermal conductivity; (**h**,**i**) Temperature dependence of lattice thermal conductivity and carrier thermal conductivity respectively; (**j**) Temperature dependence of figure of merit (ZT) [67].

**Figure 16 materials-13-05704-f016:**
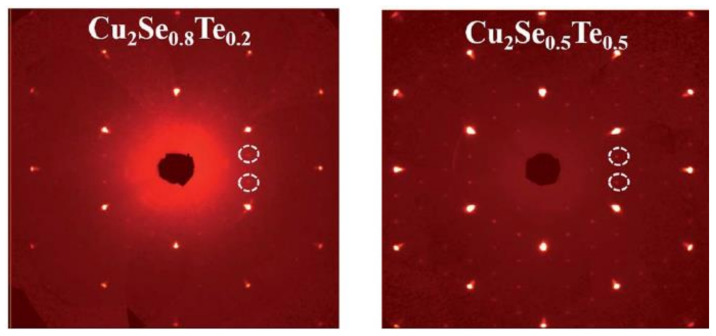
Single Crystal X-ray Diffraction (SXRD) pattern for Cu_2_Se_0.8_Te_0.2_ and Cu_2_Se_0.5_Te_0.5_, the white dashed rings are the super cell diffraction spots indicating the existence of Cu_2_Te second phase [68].

**Figure 17 materials-13-05704-f017:**
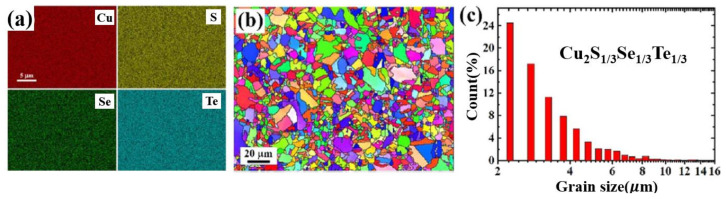
(**a**) EDS mapping for Cu_2_S_1/3_Se_1/3_Te_1/3_; (**b**) Inverse pole figure (IPF) orientation map obtained from Electron Back Scattered Diffraction image (EBSD); (**c**) Histogram of grain size distribution for the same sample in (**b**) [71].

**Figure 18 materials-13-05704-f018:**
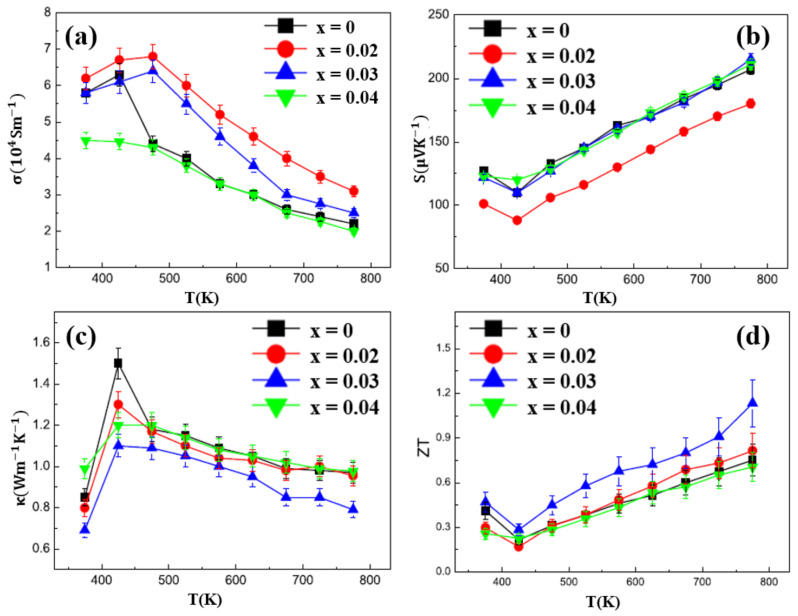
Temperature dependence of Cu_2_Se_1−x_I_x_ (x = 0–0.04). (**a**) Electrical conductivity; (**b**) Seebeck coefficient; (**c**) thermal conductivity; (**d**) Figure of merit (ZT) [74].

**Figure 19 materials-13-05704-f019:**
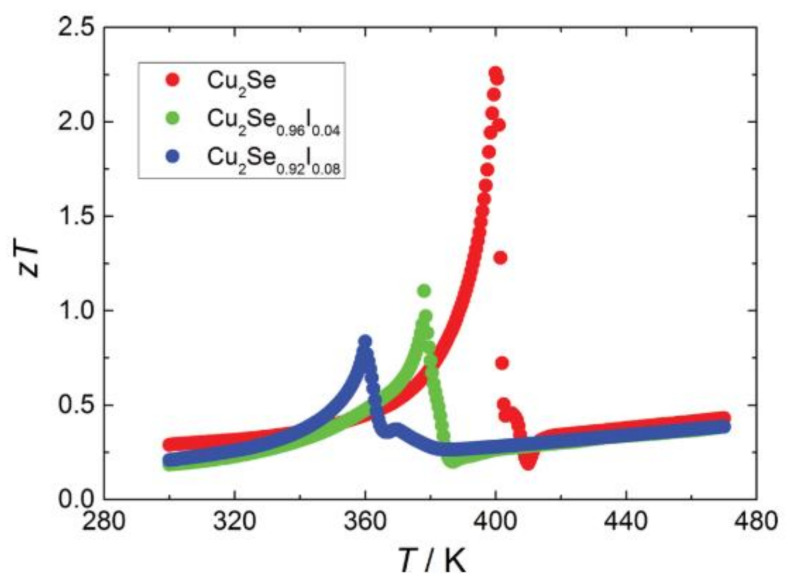
Temperature dependence of figure of merit (ZT).

**Figure 20 materials-13-05704-f020:**
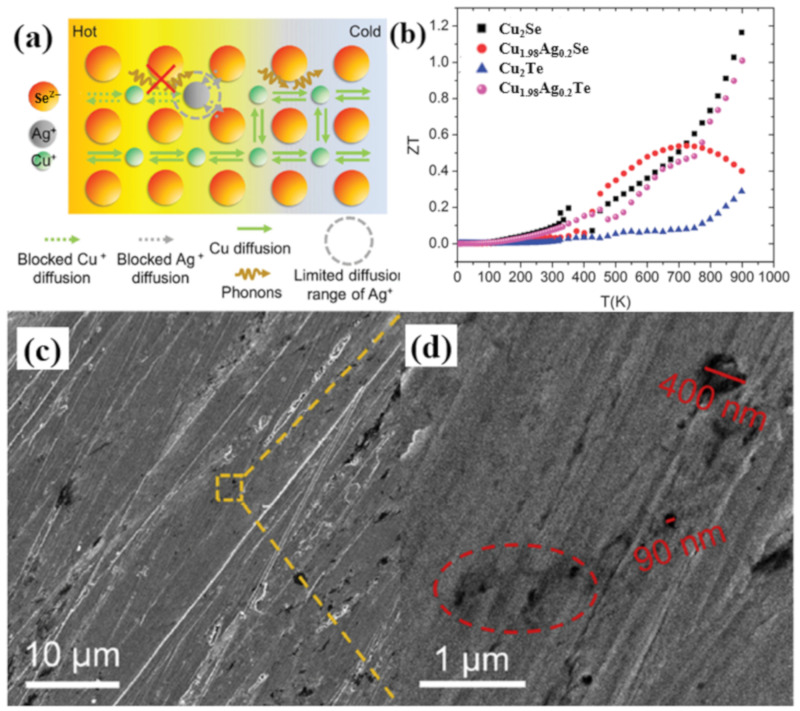
(**a**) Schematics of Cu^+^/Ag^+^ diffusion and phonon scattering. Limited diffusion range of Ag^+^ further blocks the diffusion of Cu^+^ ions and weakens the scattering of phonons by cations [78]; (**b**) Temperature dependence of figure of merit (ZT) and power factors for all sample [79]; (**c**) SEM image of the samples; (**d**) magnified image of the yellow dashed square region in (**c**), region marked in red dashed ellipses are pores form during SPS process [80].

**Table 1 materials-13-05704-t001:** Summary of best TE performances of each dopants and their synthesis process.

Group	Doping Element	Composition	Synthesis Method	TE Performance						Ref.
				*T* (K)	*S* $\left({\mu \mathrm{VK}}^{-1}\right)$	σ $\left({\mathrm{Sc}m}^{-1}\right)$	κ $\left({Wm}^{-1}K^{-1}\right)$	PF $\left({\mu \mathrm{Wm}}^{-1}K^{-2}\right)$	ZT	
**IA**	Li	Cu_1.98_Li_0.02_Se	Hydrothermal synthesis + Hot pressing	973	≈270	≈145	0.48	≈1057	$2.14{\;}^{↑↑}$	[41]
	Na	Cu_1.96_Na_0.04_Se	Hydrothermal synthesis + Hot pressing	973	≈291	≈135	≈0.52	≈1140	$2.1^{\;↑↑}$	[43]
	K	Cu_1.97_K_0.03_Se	Hydrothermal synthesis + Hot pressing	773	250	≈139	0.63	≈819	$1.19{\;}^{↑↑}$	[44]
**IIA**	Mg	Cu_1.99_Mg_0.01_Se	Single step reactive SPS	860	≈172	≈513	≈0.81	≈1511	$1.6^{\;↑}$	[48]
**IIIA**	B	0.42wt% B	Ball milling + SPS	852	245	207	~0.66	1240	$1.6{\;}^{↑↑}$	[52]
	Al	Cu_1.94_Al_0.02_Se	Melting + Ball milling + SPS	1029	≈246	261	0.611	≈1579	$2.62{\;}^{↑↑}$	[40]
	Ga	Cu_1.9925_Ga_0.0075_Se	Melting + Ball milling + SPS	823	≈188	≈293	0.6	1030	$1.4{\;}^{↑}$	[53]
	In	Cu_1.99_In_0.01_Se	Solid state transformation + Hot pressing	850	≈153	≈539	≈0.41	~1200	$2.6{\;}^{↑↑↑↑}$	[51]
**IVA**	C	0.6wt% C	Melting + Pressing	984	≈204	≈313	0.43	≈1282	$2.5^{\;↑↑↑}$	[58]
	Graphene	0.15wt% C	Ball milling + Quenched in N_2_(liq.)	870	≈174	≈278	~0.4	≈962	$2.44{\;}^{↑↑↑}$	[57]
	C-fiber	0.3wt% C	Melting + Furnance cooling	850	≈176	≈370	~0.5	≈1146	$2.4{\;}^{↑↑↑↑}$	[54]
	SiC	0.05wt% SiC	Mechanical alloying(MA) + SPS	875	≈133	≈727	≈0.59	1310	$2.0{\;}^{↑↑↑↑}$	[59]
	Sn	Cu_1.98_Sn_0.01_Se	Melting + Anealling + SPS	823	≈285	≈103	~0.69	≈829	$1.0{\;}^{↑}$	[60]
	Pb	Cu_1.95_Pb_0.015_Se	Hydrothermal synthesis + Hot pressing	973	258	≈208	0.85	≈1324	$1.52^{\;↑}$	[50]
**VA**	Bi	Cu_1.982_Bi_0.006_Se	Solvothermal method + SPS	373	≈143	≈386	~0.8	700–800	$0.43{\;}^{↑↑}$	[62]
**VIA**	S	Cu_2_Se_0.92_S_0.08_	Element melting + Anealling + SPS	1000	≈280	≈135	≈0.55	≈1052	$2.0{\;}^{↑↑}$	[63]
	Te	Cu_2_Se_0.9_tTe_0.1_	Ball milling + SPS	873	179	≈393	0.59	1255	$1.9^{\;↑↑}$	[67]
	STe	Cu_1.98_S_1/3_Se_1/3_Te_1/3_	Melting + Anealling + Sintering	1000	≈244	≈180	≈0.58	1120	$1.9^{\;↑}$	[71]
**VIIA**	Cl	Cu_2_Se_0.92_Cl_0.08_	Solid state reaction + SPS	620	≈238	≈95	≈0.59	≈533	$0.6{\;}^{↑↑↑}$	[76]
	I	Cu_2_Se_1.97_I_0.03_	Hydrothermal synthesis + Hot pressing	773	≈217	≈250	≈0.79	≈1177	$1.13{\;}^{↑↑}$	[74]
**Tran**- **sition**	Ag	Cu_1.8_Ag_0.2_Se_0.9_S_0.1_	Melting + Anealling + Hot pressing	900	≈288	≈102	≈0.48	≈829	$1.6{\;}^{↑↑↑}$	[106]
	Fe	Cu_1.99_Fe_0.01_Se	Melting + Ball milling + Quenching	823	≈159	≈454	≈0.89	≈1154	$1.07{\;}^{↑}$	[84]
	Ni	Cu_2_Se/0.05Ni_0:85_Se	Hydrothermal synthesis + Hot pressing	873	≈201	≈252	0.6	≈1014	$1.52{\;}^{↑↑}$	[83]
	Ni	Cu_1.99_Ni_0.01_Se	Melting + Ball milling + Quenching	823	≈195	≈308	0.64	1133	$1.51{\;}^{↑↑}$	[84]
	Mn	Cu_1.99_Mn_0.01_Se	Melting + Ball milling + Quenching	823	≈166	≈407	≈0.72	≈1126	$1.28^{\;↑}$	[84]
	Zn	Cu_1.99_Zn_0.01_Se	Melting + Ball milling + Quenching	823	≈173	≈389	≈0.80	1133	$1.25{\;}^{↑}$	[84]
	Sm	Cu_1.99_Sm_0.01_Se	Melting + Ball milling + Quenching	823	≈187	≈277	≈0.74	≈971	$1.07{\;}^{↑}$	[84]
	Hg	Cu_1.9_Hg_0.1_Se	Hydrothermal synthesis + Hot pressing	873	≈197	≈303	0.68	≈1165	$1.5^{\;↑↑}$	[82]

(In the table above, *T* is temperature, *S* is Seebeck coefficient, σ is electrical conductivity, κ is thermal conductivity, and PF is power factor. For numbers start with “~” means an approximate number is givien by the original article, and numbers start with “≈”, means the datum is read from the figure given by the original article using a figure digitizer. ZT values marked with uparrows have meanings as follows: ↑: increment is lower than 30% compared to the undoped sample synthesized by the authors, ↑↑: increment is higher than 30% but is lower than 100%, ↑↑↑: increment is higher than 100% but is lower than 150%, ↑↑↑↑: increment is higher than 150%.).

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
