# Peer review of "Doping Effect on Cu2Se Thermoelectric Performance: A Review"

_materials, 2020, doi:10.3390/ma13245704_

Round 1

Reviewer 1 Report

The authors summarize in a Review the Doping effects on Cu2Se with a focus on Thermoelectric performance.

The substantial review includes the results of a variety of elements in the groups I-VIIA of the Periodic Table as well as transition metals inserted into Cu2Se.  A detailed description is given with the achieved results with ZT as criteria of success.

The survey of the references is also substantial. A good basis for future work.

Author Response

Response to review report 1:

Comments

The authors summarize in a Review the Doping effects on Cu2Se with a focus on Thermoelectric performance.

The substantial review includes the results of a variety of elements in the groups I-VIIA of the Periodic Table as well as transition metals inserted into Cu2Se.  A detailed description is given with the achieved results with ZT as criteria of success.

The survey of the references is also substantial. A good basis for future work.

Response: Thanks for the reviewer’s careful reading of our manuscript and positive comments.

Reviewer 2 Report

The manuscript by Yuan-Hao Qin et al. entitled:

"Doping Effect on Cu2Se Thermoelectric Performance: A Review"

reviews the effect of various dopants on the thermoelectric performance of Cu2Se, a material that has been reported as promising for applications due to its high value of figure of merit. Albeit not being the main subject of the review, there are some points fundamental for applications that are not even mentioned in the introduction, e.g. the super-ionic transport observed in this material or the study performed by NASA on its chemical stability. Moreover, some important articles on the subject of the review are missing (e.g. those reporting the doping with Sn). Therefore, this manuscript, albeit being important and interesting, is still not suitable for publication in Materials.

Other small points:

Some figure captions do not identify for which samples the data is presented (this information is only in the text);

In Figure 6 (c) the STEM image do not corresponds to the composition maps (they correspond to the 6 (b) image);

Author Response

Response to review report 2:

Comments:

The manuscript by Yuan-Hao Qin et al. entitled:

"Doping Effect on Cu2Se Thermoelectric Performance: A Review"

reviews the effect of various dopants on the thermoelectric performance of Cu2Se, a material that has been reported as promising for applications due to its high value of figure of merit. Albeit not being the main subject of the review, there are some points fundamental for applications that are not even mentioned in the introduction, e.g. the super-ionic transport observed in this material or the study performed by NASA on its chemical stability. Moreover, some important articles on the subject of the review are missing (e.g. those reporting the doping with Sn). Therefore, this manuscript, albeit being important and interesting, is still not suitable for publication in Materials.

Other small points:

Some figure captions do not identify for which samples the data is presented (this information is only in the text);

In Figure 6 (c) the STEM image do not corresponds to the composition maps (they correspond to the 6 (b) image);

Point 1: Albeit not being the main subject of the review, there are some points fundamental for applications that are not even mentioned in the introduction, e.g. the super-ionic transport observed in this material.

Response 1: Thanks for the reviewer’s good instruction. Super-ionic transport is really important in explaining the phonon liquid behavior of Cu2Se, and we add it on Page 4, line 130-141.

Point 2: The study performed by NASA on its chemical stability is not mentioned.

Response 2: Thanks for the reviewer’s good instruction. On page 26, line 709-717, we add the chemical stability analysis of Cu2Se presented by NASA and some other groups.

Point 3: Moreover, some important articles on the subject of the review are missing (e.g. those reporting the doping with Sn).

Response 3: Thanks for the reviewer’s careful reading of our manuscript and good instruction. We are very sorry for missing one important dopant, and we have already added the discussions about Sn doped samples on page 17, line 437-450.

Point 4: Some figure captions do not identify for which samples the data is presented (this information is only in the text).

Response 4: Thanks for the reviewer’s careful reading of our manuscript and good instruction. We add the sample data for Figure.6, Figure.8, Figure.9 and Figure.20, whose sample data were lost.

Point 5: In Figure 6 (c) the STEM image does not corresponds to the composition maps (they correspond to the 6 (b) image).

Response 5: Thanks for the reviewer’s careful reading of our manuscript and good instruction. We used the original figure of the article and it seemed that the author made it wrong. We are very sorry for not noticing that, and we’ve already made the correction and rearranged the sequence of the figures.

Reviewer 3 Report

The authors present a quite extensive review paper concerning the thermoelectric performance of doped Cu2Se. After a short presentation of possible applications and concepts related to thermoelectric materials, the authors present the main results obtained in recent years which indicate improvements of thermoelectric properties of intrinsic Cu2Se. The focus is on improving the thermoelectric figure of merit by doping and related structural effects. The review presents the impact of doping with atoms from different groups (IA, IIA, IIIA, ..., transition metals etc), with a dedicated section for each.

This is a valuable contribution in the field of thermoelectrics, in particular for the newly investigated Cu2Se based compounds. However, certain issues are not well described and some others were not included, which may reduce the potential of the present review:

1. I would suggest to add more applications for thermoelectric devices, e.g. thermoelectric generators for spatial applications (where solar energy may not be available), thermocouples, waste heat in automotive industry etc.

2. Concerning Eqs. (1-5), one should indicate in more detail the conditions and approximations for which these are valid.

3. Exploiting the unique features of low dimensional systems (e.g. nanowires) has not been properly addressed.

4. Where it is possible, the authors should also include discussions regarding theoretical modeling of these compounds (ab initio calculations in particular).

5. The English should be re-checked throughout the manuscript. For instance,
"It is known that only when ZT reaches 3, which is about half of the Carnot efficiency[11], and η = 20% − 25% the thermoelectric device can be commercially available[12]."
I understand that the authors meant that an efficient thermoeletric device should have ZT larger than 3. The meaning is that the thermoelectric device can be made commercially feasible.

Other statements which need rephrasing and the list is not exhaustive:

"However, _even has_ a relatively..."
"highly aligned Cu+ ion layers _coexisted_ with aligned lamellae, ..."

Author Response

Response to review report 3:

Comments:

The authors present a quite extensive review paper concerning the thermoelectric performance of doped Cu2Se. After a short presentation of possible applications and concepts related to thermoelectric materials, the authors present the main results obtained in recent years which indicate improvements of thermoelectric properties of intrinsic Cu2Se. The focus is on improving the thermoelectric figure of merit by doping and related structural effects. The review presents the impact of doping with atoms from different groups (IA, IIA, IIIA, ..., transition metals etc), with a dedicated section for each.

This is a valuable contribution in the field of thermoelectrics, in particular for the newly investigated Cu2Se based compounds.

Point 1: I would suggest to add more applications for thermoelectric devices, e.g. thermoelectric generators for spatial applications (where solar energy may not be available), thermocouples, waste heat in automotive industry etc.

Response 1: Thanks for the reviewer’s careful reading of our manuscript and good instruction. The applications of TE materials has been added on page 3, line 99-106.

Point 2: Concerning Eqs. (1-5), one should indicate in more detail the conditions and approximations for which these are valid.

Response 2: Thanks for the reviewer’s careful reading of our manuscript and good instruction. Detailed discussion about equation (1-5) (the approximations that is used in those equations and conditions they are valid) has been added on page 2, line 49-74.

Point 3: Exploiting the unique features of low dimensional systems (e.g. nanowires) has not been properly addressed.

Response 3: Thanks for the reviewer’s careful reading of our manuscript and good instruction. Discussions on low dimensional systems have been added on page 26, line 701-709.

Point 4: Where it is possible, the authors should also include discussions regarding theoretical modeling of these compounds (ab initio calculations in particular).

Response 4: Thanks for the reviewer’s careful reading of our manuscript and good instruction. Discussions on ab initio calculations have been added on page 4, line 113-117 and line 130-141.

Point 5: The English should be re-checked throughout the manuscript. For instance,
"It is known that only when ZT reaches 3, which is about half of the Carnot efficiency[11], and η = 20% − 25% the thermoelectric device can be commercially available[12]."
I understand that the authors meant that an efficient thermoeletric device should have ZT larger than 3. The meaning is that the thermoelectric device can be made commercially feasible.

Other statements which need rephrasing and the list is not exhaustive:

"However, _even has_ a relatively..."
"highly aligned Cu+ ion layers _coexisted_ with aligned lamellae, ..."

Response 5: Thanks for the reviewer’s careful reading of our manuscript. After further checked by several authors, many typographical and grammar mistakes were corrected. Those correction for mistakes listed above are on page 2 line 79-81, page 4 line 149 and page 8 line 251.

Reviewer 4 Report

This review on Cu2Se is well written and detailed. Cu2Se is a high ZT materials but his stability is quite poor so I would have like to see more detailed discussion on this matter.

the paper could be publish after minor editing. Especially many formula need to be edited, especially subscript and superscript. Please carefully proofread. 

Page 5, you need to correct "Sink Plasma Sintering" by "Spark Plasma Sintering"

References authors names need to check carefully. For exemples p9 Olevera et al. but Olvera in reference 38.

Sometimes only given name is use (Tristan et al, instead of Tristan W. Day et al. for example p. 22) sometimes only family name, sometimes both. Please correct homogenize this.

figures quality can be improved, some are difficult to read.

Author Response

Response to review report 4:

Comments:

This review on Cu2Se is well written and detailed.

Points 1: Cu2Se is a high ZT materials but his stability is quite poor so I would have like to see more detailed discussion on this matter.

Response 1: Thanks for the reviewer’s careful reading of our manuscript and good instruction. We add the chemical stability analysis of Cu2Se presented by NASA and some other groups on page 26 line 709-717.

Point 2: the paper could be published after minor editing. Especially many formula need to be edited, especially subscript and superscript. Please carefully proofread. 

Response 2: Thanks for the reviewer’s careful reading. After further checked by several authors, many formulas have been corrected, which are on line 19, 55, 85, 88, 103, 501, 547, 605-609, 619, 636, 677, 691, 694

Point 3: Page 5, you need to correct "Sink Plasma Sintering" by "Spark Plasma Sintering"

Response 3: Thanks for the reviewer’s careful reading. It has been corrected on page 6, line 205

Point 4: References authors names need to check carefully. For exemples p9 Olevera et al. but Olvera in reference 38.

Sometimes only given name is use (Tristan et al, instead of Tristan W. Day et al. for example p. 22) sometimes only family name, sometimes both. Please correct homogenize this.

Response 4: Thanks for the reviewer’s careful reading and good instruction. All references authors’ names have been transformed into full names. Corrections are one page 10 line 283, page 19 line 501, page 23 line 593.

Point 5: figures quality can be improved, some are difficult to read.

Response 5: Thanks for the reviewer’s careful reading and good instruction. We improved the figure quality as best as we can, and some figures in the original article is of low quality and no figure of higher quality is presented on their website. Quality of Fig.6, Fig.7, Fig.15, Figure.20 has been improved.

Round 2

Reviewer 2 Report

Most of the Reviewer’s questions were answered and the manuscript is ready to be accepted for publication after the correction of some minor English errors.

Author Response

Thanks for the reviewer’s careful reading of our manuscript and positive comments. We have editing our manuscript as best as we can. All English errores we found have been corrected and changes have been made to make our manuscriped more precise and readable. Those changes and corrections can be found in line 24-25, 31, 37, 122, 167, 174, 176, 218, 224, 303, 319, 418, 428-429, 526, 541, 637, 645, and 653.